# Female-specific myoinhibitory peptide neurons regulate mating receptivity in *Drosophila melanogaster*

Yong-Hoon Jang [1], Hyo-Seok Chae[1] & Young-Joon Kim[1]

Upon mating, fruit fly females become refractory to further mating for several days. An ejaculate protein called sex peptide (SP) acts on uterine neurons to trigger this behavioural change, but it is still unclear how the SP signal modifies the mating decision. Here we describe two groups of female-specific local interneurons that are important for this process —the ventral abdominal lateral (*vAL*) and ventral abdominal medial (*vAM*) interneurons. Both *vAL* and *vAM* express myoinhibitory peptide (*Mip*)-GAL4. *vAL* is positive for Mip neuropeptides and the sex-determining transcriptional factor *doublesex*. Silencing the *Mip* neurons in females induces active rejection of male courtship attempts, whereas activation of the *Mip* neurons makes even mated females receptive to re-mating. *vAL* and *vAM* are located in the abdominal ganglion (AG) where they relay the SP signal to other AG neurons that project to the brain. Mip neuropeptides appear to promote mating receptivity both in virgins and mated females, although it is dispensable for normal mating in virgin females.

[1] School of Life Sciences, Gwangju Institute of Science and Technology, Gwangju 61005, Republic of Korea. Correspondence and requests for materials should be addressed to Y.-J.K. (email: kimyj@gist.ac.kr)

Depending on environmental and physiological conditions, animals often need to respond differently to the same stimulus. Such behavioural switching is made possible by functional reconfigurations of existing neural circuits. Even the circuits that drive innate behaviours can undergo such reconfigurations[1]. The postmating responses (PMR) of female *Drosophila melanogaster* are a clear example of this[2]. Prior to mating, female *Drosophila* accept the courtship advances of males. After mating, however, they actively reject courtship. This suggests the neural circuits that regulate female mating receptivity can exist in at least two distinct functional states, receptive and non-receptive. PMR have emerged as an important behavioural model in which to study the neural processes that reconfigure the functional state of neural circuits in a stable but reversible manner[3].

PMR is induced chemically by a male seminal protein called sex peptide (SP)[4,5]. During copulation, the male transfers SP along with sperm into the female uterus[6,7]. SP activates SP receptor (SPR) in a small number of SPR-positive sensory neurons (SPSNs), which innervate the uterine lumen and send afferent processes into the tip of the abdominal ganglion (AG)[8–11]. After the SPSNs, the SP signal is relayed to higher brain areas by two or three SP abdominal ganglion (SAG) neurons, which extensively innervate the AG and project into the dorsal protocerebrum[12]. The SP and SPR pathway seems inhibitory and it signals the postmating state by silencing the SPSNs and SAG neurons[9,12]. Silencing the SPSNs or SAGs renders the virgin females unreceptive to mating, while forced activation of the SAGs overrides the SPSN silencing and makes the females regain mating receptivity[12]. Although the SPSN–SAG circuit explains the route by which the peripheral SP signal enters the central nervous system (CNS), the neural and molecular mechanisms that reconfigure the state of the brain and maintain receptivity remain unclear[13].

Neuromodulators like monoamines and neuropeptides change the properties of neural circuits by altering neuronal excitability through slow-acting metabotropic receptors[14]. Decision-making circuits and pattern generators are often multifunctional, with their properties being reconfigured by neuromodulators[15]. In the mammalian thalamus, for example, a cocktail of modulatory substances mediates a switch in the firing pattern of neurons associated with the sleep–wake transition[16]. Likewise, we hypothesize that modulatory substances reconfigure the female mating circuits. Several neuromodulators have already been linked to reproductive behaviours in insects. For example, octopamine has long been implicated in sperm storage, mating and egg laying[17–21]. Neurons expressing the neuropeptide SIFamide are associated with female hyper-receptivity[22]. Still, the roles neuromodulators and the neurons that produce them play in female mating behaviours remain largely unexplored[23]. Here we identify two groups of female-specific local interneurons that relay the SP signal from the SPSNs to the SAG neurons and show that a neuropeptide myoinhibitory peptide (Mip) expressed in some of these neurons promotes mating receptivity in females.

## Results

### Screen for postmating switch neurons.
To identify CNS neurons that modulate female mating decisions, we performed a genetic screen to acutely manipulate the activity of groups of specific neuromodulator neurons in the fly brain and examine the effects on PMR. We used a panel of neuromodulator-specific GAL4 lines to drive the expression of *Drosophila* transient receptor potential A1 (dTrpA1) or temperature-sensitive shibire (Shi[ts]) to temporarily activate or silence (respectively) groups of neurons with a simple temperature shift[24,25]. For behavioural assays, we mated individual virgin females carrying either one or two transgenes

(GAL4 and UAS) to a naive *Canton-S* (CS) male and then tested for receptivity with a second naive CS male 48 h later (re-mating assay). We induced neuronal silencing or activation by shifting the flies to the restrictive temperature (30 °C) shortly after they mated with the first male (Supplementary Fig. 1a). Of the 39 GAL4 lines we screened, 5 exhibited substantial differences when tested at restrictive vs. permissive temperatures (Supplementary Fig. 1a). However, of these five, only two lines—*Mip-GAL4* line and the Diuretic hormone 44 (*Dh44*)-*GAL4* line—continued to exhibit significant differences when compared to the appropriate genetic controls (Supplementary Fig. 1b–f). Here we focused our analysis on *Mip-GAL4*, which produces the strongest mating and re-mating phenotypes.

### *Mip-GAL4* neurons switch the mating decision.
Previously, we reported the expression of *Mip-GAL4* in most Mip-positive neurons in the CNS that are labelled by either an anti-Mip antibody or Mip mRNA in situ hybridization[26]. We will, therefore, refer to *Mip-GAL4* neurons as *Mip* neurons. In our initial screen, we activated or silenced neurons for ~ 48 h prior to the re-mating assay. To avoid any unexpected anomalies caused by chronic (~ 48 h) neuronal activation or silencing, we shifted females to the restrictive temperature 30 min prior to the re-mating assay (Fig. 1a). Since *Mip* neuron activation increases mated female receptivity, we asked whether *Mip* neuron silencing suppresses virgin female receptivity. This was indeed the case— we found that >80% of females with silenced *Mip* neurons fail to mate within 1 h (*Mip* > *Shi[ts]* 30 °C; Fig. 1b). Next we examined the receptivity of virgin females with activated *Mip* neurons (*Mip* > *dTrpA1*, 30 °C). Unexpectedly, *Mip* neuron activation also moderately suppresses virgin female receptivity (Fig. 1c). The fact that *Mip* neuron silencing and activation can both reduce mating receptivity depending on mating status suggests that *Mip* neurons comprise at least two functionally distinct populations, one that facilitates mating and another that suppresses it. To separate these populations genetically, we prepared additional *GAL4* transgenes with various fragments of the 5′-upstream regulatory regions of *Mip* and screened the resulting flies for lines that recapitulate the mating or/and re-mating phenotypes of *Mip-GAL4* neuron activation (Supplementary Fig. 2a–c). None of the *GAL4* lines we tested were able to influence the re-mating of mated females, but thermal activation with *Mip[6.0]-GAL4* suppresses the mating receptivity of virgin females. *Mip[6.0]-GAL4* carries a 6.0 kb fragment of the 5′-upstream regulatory region, whereas *Mip-GAL4* carries 7.2 kb fragment (Supplementary Fig. 2a). To determine whether *Mip[6.0]-GAL4* targets a subset of the *Mip-GAL4* neurons responsible for inducing mating refractoriness in virgin females, we generated a *Mip[6.0]-GAL80* line and combined it with *Mip-GAL4*. In these experiments, GAL80 restricts GAL4 activity to a subset of the *Mip* neurons. Indeed, *Mip[6.0]-GAL80* seems to be expressed in the neurons that when activated suppress mating receptivity, because at 30 °C, *Mip[6.0]-GAL80* blocks the suppression of receptivity observed in *Mip* > *dTrpA1* virgin females but not in *Mip* > *Shi[ts]* females (Fig. 1d, e). Mated flies, however, re-mate more frequently than those lacking *Mip[6.0]-GAL80* (Fig. 1g, i). *Mip[6.0]-GAL80* does not affect the phenotypes induced by *Mip* neuron silencing (Fig. 1f, h).

Mated females actively reject courting males by forcefully extruding their ovipositors in response to male advances. We examined the effects of *Mip* neuron activation or silencing on the rejection behaviour while subtracting the *Mip[6.0]-GAL80* neurons. We found *Mip* neuron activation suppresses ovipositor extrusion by 80% (Fig. 1k), whereas *Mip* neuron silencing enhances it (Fig. 1j and Supplementary Movie 1). To exclude the possibility that the *Mip* neurons are involved in the motor output required for

ovipositor extrusion rather than controlling mating receptivity, we asked whether *Mip* neuron silencing induces ovipositor extrusion even in the absence of courting males (Fig. 1l). In *Drosophila*, the *fruitless* (*fru*) gene encodes a sexually dimorphic transcription factor that is closely linked to male sexual orientation[27]. Males homozygous for *fru^F* show very little courtship towards females[28]. When paired with *fru^F*/*fru^F* males, virgin females with silenced *Mip* neurons show no increase in ovipositor extrusion, indicating that the rejection behaviour caused by *Mip* neuron silencing occurs only in response to male advances. Indeed, ovipositor extrusion never occurs in isolated females. It is important to note that mated control females show

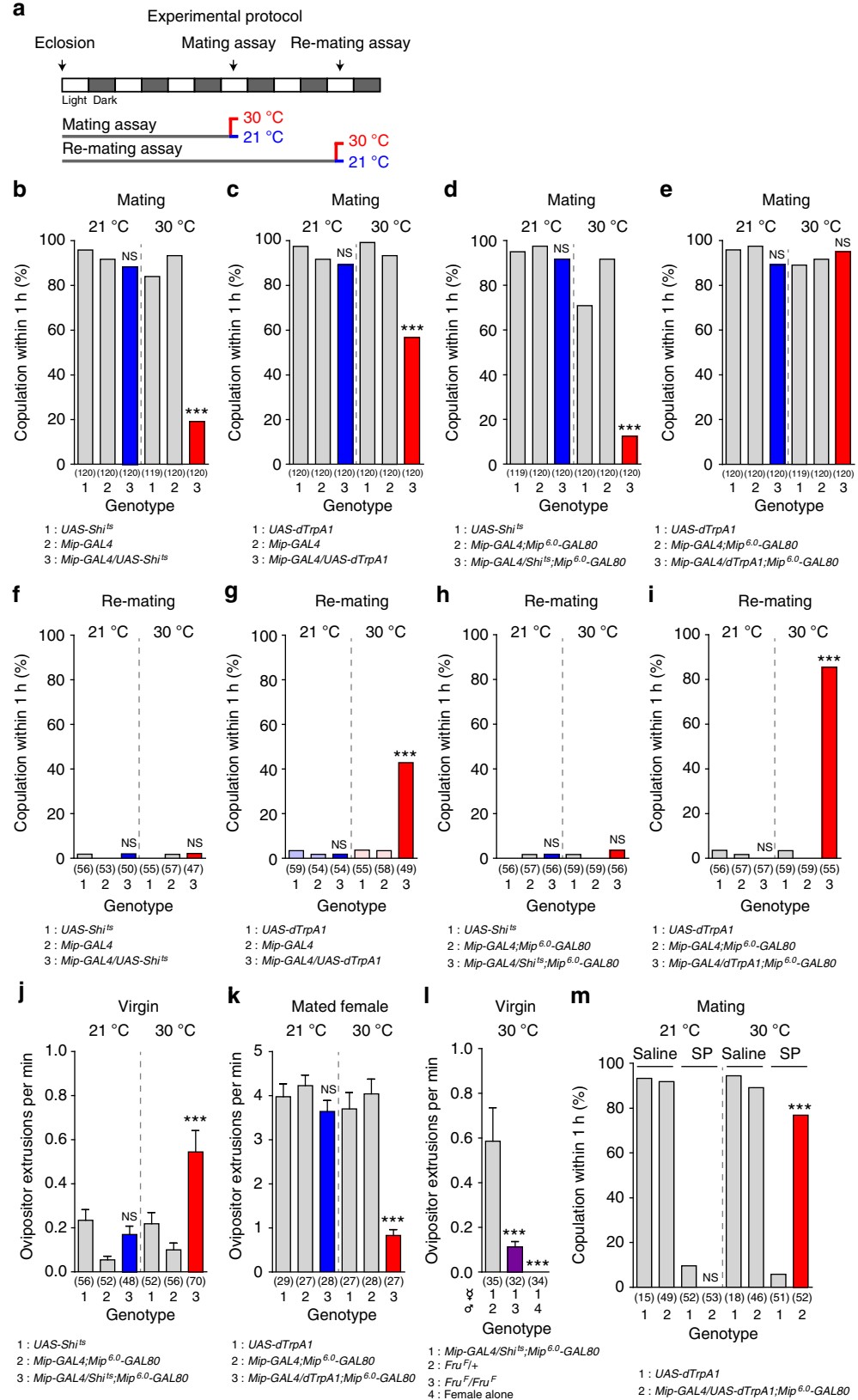

~ 8 times more frequent ovipositor extrusion than virgin females with silenced *Mip* neurons. This may be due to insufficient *Mip* neuron silencing or it may indicate additional neural components operating in mated females. Nevertheless, these results suggest the *Mip* neurons that are not labelled by *Mip6.0-GAL80* can switch the mating decision on or off regardless of a female's previous mating experience. Despite robust effects on receptivity, neither activation nor silencing of *Mip* neurons affects the fecundity of mated females (Supplementary Fig. 3a, b). In virgin females, however, *Mip* neuron silencing slightly increases egg laying (Supplementary Fig. 3c). This is similar to the increase in virgin egg laying induced by silencing the SAG neurons[12].

***Mip-GAL4* neurons signal downstream of SP**. The effects of SP on mating receptivity and egg laying are mediated by SPR expressed in a small subset of uterine *fruitless* (*fru*), *doublesex* (*dsx*) and *pickpocket* (*ppk*)-positive SPSNs that arborize in the lumen of the uterus and project to the CNS[9–11]. Their innervation pattern suggests that these neurons detect SP inside the uterus and relay this signal directly into the CNS. We examined female flies expressing dsRed in *Mip-GAL4* neurons and enhanced green fluorescent protein (EGFP) in *ppk* neurons to determine whether SPSNs express *Mip-GAL4*. Because there was no *Mip-GAL4*-positive process among *ppk*+ SPSNs in the uterus (arrows in Supplementary Fig. 4a), we explored the possibility that SP directly modulates central *Mip* neurons. This is important because male-derived SP circulates in the haemolymph of mated females[6, 29] and SPR is expressed broadly in the CNS[8]. Although *Mip* neuron-specific knockdown of SPR does not affect PMR, *ppk* neuron-specific *SPR* knockdown strongly suppresses PMR, allowing flies to re-mate[9, 10] (Supplementary Fig. 4b). Furthermore, SPR expression in *ppk* neurons but not *Mip* neurons rescues PMR in SPR mutants (Supplementary Fig. 4c). These results indicate that SPR expression in *Mip* neurons is neither necessary nor sufficient to elicit PMR.

We next examined the behaviour of flies expressing a membrane-tethered form of SP (mSP)[30] in *Mip* neurons to test the possibility that SP directly modulates *Mip* neurons via SPR or a second unidentified SP receptor. As previously reported[9], virgin females expressing mSP in their *ppk* neurons are as unreceptive as mated females, but females expressing mSP in *Mip* neurons are as receptive as wild-type controls (Supplementary Fig. 4d). We conclude, therefore, that SP does not directly modulate *Mip* neurons and that *Mip* neurons constitute a CNS circuit that operates downstream of SP-SPR signalling. To test this, we asked whether *Mip* neuron activation can override the effects of SP *in vivo*. We injected virgin females with synthetic SP, which suppresses mating receptivity[4, 8, 31]. Thermal activation of *Mip* neurons makes these SP-injected females as receptive as saline-injected controls (Fig. 1m). This epistatic effect of *Mip* neuron activation over SP injection indicates that the *Mip* neurons function downstream of SP-SPR signalling, presumably as SPSNs.

***Mip* neurons in the CNS**. Since *Mip* neurons appear to function downstream of SPSNs, and SPSNs send afferent processes into the AG, we reasoned that the behaviourally relevant *Mip* neurons

reside in the CNS. Thus we examined the expression pattern of *Mip-GAL4*, *Mip6.0-GAL4* and *Mip6.0-GAL80* in the female CNS (Fig. 2a–c) using anti-Mip staining as a reference. Although the CNS expression of *Mip-GAL4* was previously reported[26], we included it in our analysis for direct comparisons with other transgenes (Fig. 2a). As shown previously, *Mip-GAL4* is expressed in ~ 230 CNS neurons. Virtually all 52 anti-Mip neurons are positive for *Mip-GAL4*, but the converse is not true. Likewise, *Mip6.0-GAL4* is expressed in most anti-Mip neurons, except for two pairs of superior anterior medial (SAM) neurons and one pair of inferior contralateral interneurons in the brain (in dotted red circles in the cartoon in Fig. 2b). *Mip6.0-GAL4* is also expressed in additional neurons lacking anti-Mip staining, which include myriad optic lobe neurons (indicated by green shading in the cartoon in Fig. 2b), 112 non-optic lobe brain neurons and 56 ventral nerve cord (VNC) neurons. Except for 46 anti-Mip neurons that express both *Mip-GAL4* and *Mip6.0-GAL4* (compare cartoons in Fig. 2a, b), the both *GAL4* neuron populations seem largely different from each other. Lastly, we examined the *Mip6.0-GAL80* expression using females carrying both the *Mip-Gal4* and *Mip6.0-GAL80* transgenes. Despite the overlap of *Mip-GAL4* and *Mip6.0-GAL4* expression in anti-Mip neurons, *Mip6.0-GAL80* blocks the *Mip-GAL4* expression in few *Mip6.0-GAL4*-positive brain anti-Mip neurons including the SAM, antennal lobe superior, central anterior and inferior anterior medial neurons (see dotted red circles in the cartoon in Fig. 2c). Although all anti-Mip neurons in the VNC are positive for both *Mip-GAL4* and *Mip6.0-GAL4*, *Mip6.0-GAL80* does not block GAL4 activity in any of the VNC neurons.

**Optimization of dTrpA1-mediated *Mip* neuron activation**. The population of *Mip-GAL4* neurons seems to be composed of at least two functionally opposing subsets: dTrpA1-mediated activation of one subset suppresses mating, while activation of the other promotes mating. We suspected that these subsets may have different temperature requirements for thermal activation. If so, we hoped to be able to activate one subset without activating the other simply by optimizing the activation temperature. Indeed, incubation at 27 °C does not suppress the virgin mating receptivity of *Mip > dTrpA1* females, but still increases re-mating by ~ 40% (Fig. 2d, e). Consistent with this result, *Mip6.0-GAL4* neurons expressing dTrpA1 have little impact on mating receptivity at 27 °C but almost completely suppress mating receptivity at 30 °C. Activation of the mating promoting subset alone (*Mip-GAL4*, *Mip6.0-GAL80*, *UAS-dTrpA1*) increases re-mating by ~ 40% at 27 °C and almost fully at 30 °C. Together, these observations indicate that a 27 °C incubation at least partially activates the pro-mating circuit without activating the antimating circuit in *Mip > dTrpA1* females.

***Mip* is involved in female mating receptivity**. We previously generated a Mip-null allele (*Mip1*) in which the entire *Mip* coding sequence is replaced by the *mini-white* gene[26]. When paired with naïve *CS* males, Mip-deficient virgin females (*Mip1/1*) mate normally (Fig. 3a). We previously observed that knockdown of *Mip* across the nervous system does not affect the receptivity of either

**Fig. 1** *Mip-GAL4* neurons switch the mating decision. **a** The experimental procedure used for the behaviour assays. Female flies were incubated at the indicated temperatures for 30 min prior to the mating or re-mating assays. For the re-mating assay, females were mated individually with naïve *CS* males and kept at 23 °C for 48 h before being paired with a second naïve *CS* male. **b–e** Mating frequencies of virgin females of the indicated genotypes, scored as the percentage of females that copulate within 1 h. **f–i** Re-mating frequencies of mated females. **j–l** Ovipositor extrusions per minute observed in virgin females **j**, **l** or mated females **k** paired with naïve *CS* males **j**, **k**, males of the indicated genotype or no male **l**. **m** Mating frequencies of virgin females of the indicated genotypes injected with 12 pmol of sex peptide (SP) in saline or saline alone 2 h prior to the assay. The numbers in parentheses represent *n*. NS indicates non-significance (*P* > 0.05); \*\*\**P* < 0.001 for comparisons against both controls (grey bars); Chi-square test **b–i**, **m** or one-way ANOVA with Tukey's multiple comparison test **j–l**. Error bars in **j–l** indicate s.e.m.

virgin or mated females[32]. Next, we examined re-mating of mated $Mip^{1/1}$ females while activating their $Mip$ neurons. Thermal activation of $Mip$ neurons does increase re-mating in the $Mip$ mutant background but not nearly as much as in the presence of the wild-type $Mip$ allele (Fig. 3b). Although this experiment indicates that Mip promotes re-mating, neither Mip

overexpression in the $Mip$ neurons ($Mip > Mip$) nor in the entire nervous system ($elav > Mip$) affects mating latency in virgins or re-mating frequency in mated females (Fig. 3c, d, respectively).

The $Mip$ gene encodes several closely related Mip neuropeptides, which are highly selective and potent agonists of SPR[32, 33]. Thus it seems likely that $Mip$ neuron activation induces the

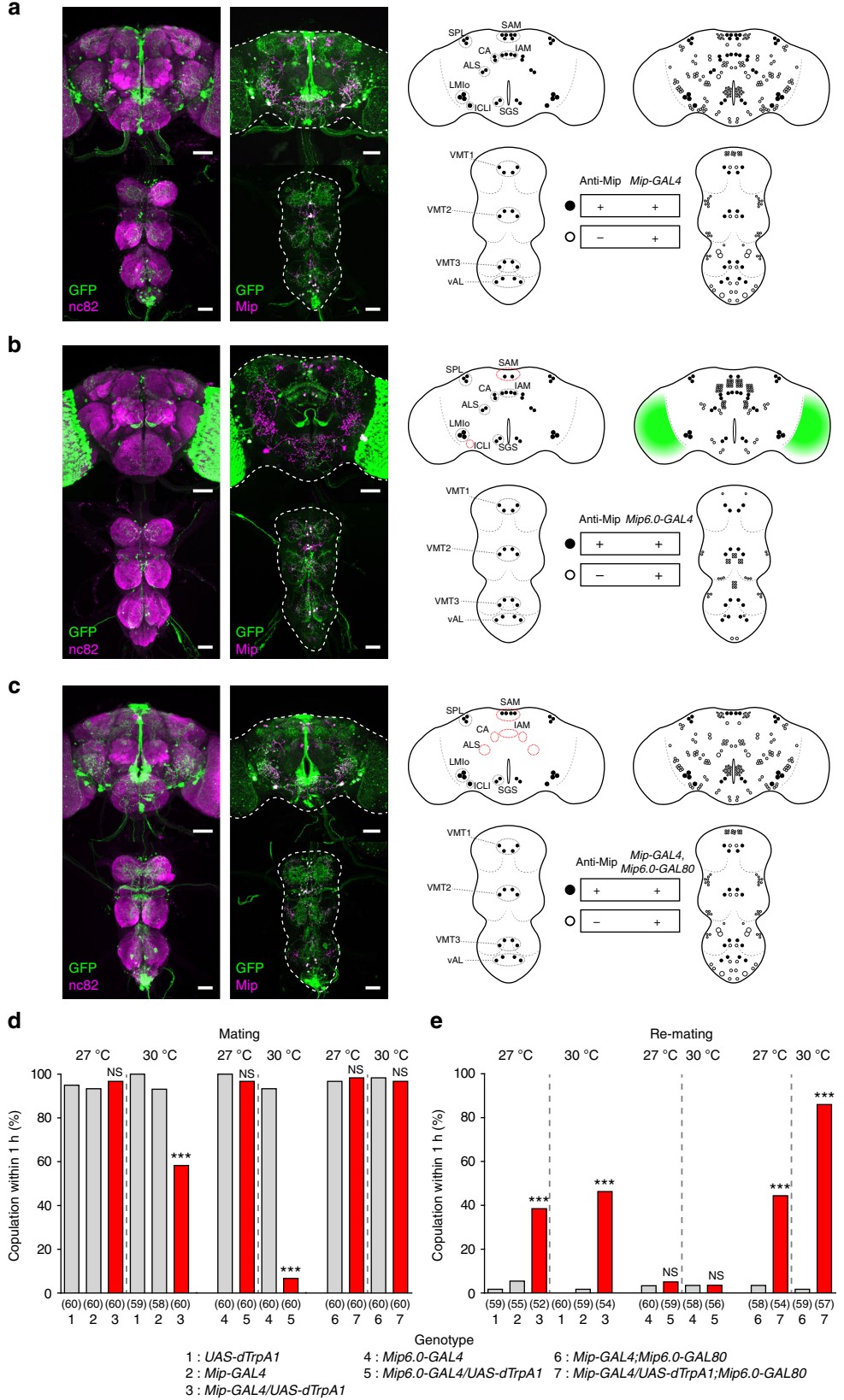

secretion of Mip peptides, which, in turn, activate SPR and suppress mating receptivity. Unexpectedly, however, we found that *Mip* neuron activation increases the sexual receptiveness of mated females, partly through Mip gene products (Fig. 3b). It, therefore, seems unlikely that Mip acts through SPR to promote re-mating. To test this directly, we wanted to examine re-mating in SPR-deficient females while activating their *Mip* neurons. Unfortunately, this experiment is impossible; SPR-deficient females are sexually receptive regardless of mating status. Instead, we asked whether *Mip* neuron activation facilitates the first mating event and then whether its enhancement of mating persists in the absence of Mip or SPR (Fig. 3e, f, respectively). We first compared the cumulative mating of *Mip > dTrpA1* virgin females paired with *CS* males at 21 and 27 °C. We used 27 °C for thermal activation to minimize the activation of the subpopulation of *Mip* neurons that inhibits mating (i.e., those that express *Mip*[6.0]*-GAL4*, see Fig. 2d, e). Virgin females whose *Mip* neurons are activated at 27 °C engage in their first mating more rapidly than controls incubated at 21 °C (Fig. 3e). Subsequently, we examined *Mip > dTrpA1* and its controls in the *Mip-* and *SPR-* deficient backgrounds. As expected, *Mip* neuron activation does not promote mating in females that lack Mip (Fig. 3e). Surprisingly, the loss of SPR does not affect the increase in mating induced by *Mip* neuron activation (Fig. 3f). These results strongly suggest that the effects of Mip on mating are mediated by a receptor other than SPR.

**Dsx⁺ abdominal *Mip* neurons control female receptivity.** *Mip-GAL4* is expressed mainly in the central neurons of the brain and VNC[26]. To map the *Mip* neurons that are functionally relevant to mating behaviour, we examined the *Mip* neurons that express *fruitless* (*fru*) or *doublesex* (*dsx*) because these genes are causally linked to several gender-specific behaviours[21, 34–38]. We previously reported that a small number of brain *Mip* neurons are positive for *fru*[FLP] (Supplementary Fig. 5a)[26]. Neither silencing nor activating these *Mip* and *fru*[FLP] double-positive central neurons affects female mating receptivity (Supplementary Fig. 5b, c). In contrast, silencing *Mip* and *dsx*[FLP] double-positive neurons (hereafter *Mip/dsx*) markedly suppresses virgin receptivity (Fig. 4a), although their thermal activation does not increase re-mating in mated females (Fig. 4b). We next examined EGFP-labelled *Mip/dsx* neurons in the CNS of females carrying *Mip-GAL4, UAS-FRT-stop cassette-FRT-mCD8-EGFP* and either one or two copies of the *dsx*[FLP] transgene (Fig. 4c, d). After noting more reproducible labelling of *Mip/dsx* neurons in females carrying two copies of *dsx*[FLP], we detected two pairs of *Mip/dsx* neurons in the brain and two pairs in the AG (Fig. 4c, d). To separate the functions of these neurons, we added an *otd*[FLP] transgene, which is active exclusively in the brain, not the VNC (Supplementary Fig. 5d)[39]. Neither silencing nor activating the *Mip/otd* neurons affects mating in virgin or mated females (Supplementary Fig. 5e, f). This suggests it is unlikely that the brain *Mip* neurons are involved. Thus we concluded that *Mip/dsx*

neurons in the AG constitute a part of the neural circuit that, when active, maintains female sexual receptivity. The *Mip/dsx* neurons in the AG are also stained by the anti-Mip antibody (Fig. 4e, upper panel). Based on the location of their somas, we propose naming them ventral anterior lateral neurons of the AG (hereafter *vAL*). The anatomy of the *vAL* neurons suggests they are local interneurons that extensively innervate the AG and project to the meso-thoracic ganglion along the midline (Fig. 4e, lower panel). We next defined the input and output domains of *vAL* by driving expression of the postsynaptic marker nSyb and the presynaptic marker Dscam, respectively. We found that the *vAL* neurons appear to receive inputs from the abdomen and send outputs to both the abdomen and thorax via processes that run along the midline (Fig. 4f, g).

**vAL neurons intermingle with SPSN and SAG neurons.** The SPSN neurons that detect SP in male ejaculate send their axon terminals to the tip of the AG. Because *vAL* neurons are the only Mip-positive neurons in the AG and because the anti-Mip staining in the AG co-localizes with both nSyb and Dscam (Fig. 4h, i), we suspected we could use anti-Mip staining of SPSN neural processes in the AG to visualize anatomical interactions between *vAL* and SPSNs. As expected, we observed intermingling of anti-Mip staining with SPSN processes in the AG (Fig. 4j). Likewise, we also noted anatomical interactions between *vAL* neurons and the SAG neurons that project to higher brain areas (Fig. 4k)[12].

**Female-specific *Mip* neurons in the AG.** Successful mating requires gender-specific neural processing. Thus we compared *Mip* neurons in males and females. Since the brain *Mip* neurons are likely irrelevant, we focussed our analysis on the VNC neurons. We found clear sexual dimorphism in the *Mip* neurons of the AG. The abdominal *Mip* neurons labelled with *Mip-GAL4*, *Mip*[6.0]*-GAL80* and *UAS-mCD8-EGFP* in females are divided into five groups according to the relative locations of their somas (Fig. 5a): the anti-Mip and *dsx*-positive *vAL*, ventral anterior medial (*vAM*), small ventral posterior medial (*s-vPM*), large ventral posterior medial (*l-vPM*), and small medial posterior medial (*s-mPM*). Apart from *vAL*, no other AG cells express anti-Mip. In males, we found no labelled neurons in the AG regions where *vAL* and *vAM* neurons are in females (Fig. 5a, b).

Because *Mip* neuron activation increases re-mating, we reasoned that the relevant *Mip* neurons would show higher activity levels in virgin females than mated females. We, therefore, asked whether AG neurons change their activity before and after mating. We monitored *Mip* neuron activity using an end-point Ca²⁺ reporting system, the transcriptional reporter of intracellular Ca²⁺ (TRIC). TRIC is designed to increase GFP expression in proportion to intracellular Ca²⁺ levels[40]. In virgin females, we detected robust TRIC signal in *vAM*, *l-vPM* and SAG neurons (Fig. 5c). We measured the SAG neurons as a positive control. Measuring 48 h after mating, we observed significant

**Fig. 2** Mip neurons in the CNS. **a–c** *Mip* neurons in a 4-day-old virgin female bearing **a** *Mip-GAL4, UAS-mCD8-EGFP*, **b** *Mip*[6.0]*-GAL4, UAS-mCD8-EGFP* and **c** *Mip-GAL4, Mip6.0-GAL80, UAS-mCD8-EGFP* stained with anti-GFP (green) and anti-nc82 (magenta) or anti-GFP (*green*) and anti-Mip (magenta) antibodies. The left schematic shows anti-Mip and EGFP-positive neurons (closed circles). The right schematic includes EGFP neurons that lack anti-Mip labelling (open circles). In the left schematic, dotted black circles group anti-Mip neuron classes. Dotted red circles indicate groups that lack complete labelling of *Mip* neuron(s), compared with *Mip-GAL4*. Green shading in the schematic in **b** indicates a large number of uncounted EGFP-positive cell bodies. *SAM* superior anterior medial, *SPL* superior posterior lateral, *IAM* inferior anterior medial, *ALS* antennal lobe superior, *SGS* subesophageal superior, *CA* central anterior, *ICLI* inferior contralateral interneurons, *SGI* subesophageal inferior, *LMlo* lateral MIP-IR optic lobe, *VMT1* ventral medial thoracic 1, *VMT2* ventral medial thoracic 2, *VMT3* ventral medial thoracic 3, *vAL* ventral lateral abdominal. Scale bars, 50 μm. **d** Mating frequencies of virgin females of the indicated genotypes, scored as the percentage of females that copulate within 1 h. **e** Re-mating frequencies of mated females. The numbers in parentheses indicate *n*. NS indicates non-significance (*P* > 0.05); ***P < 0.001 for comparisons against both controls (grey bars); Chi-square test **d**, **e**

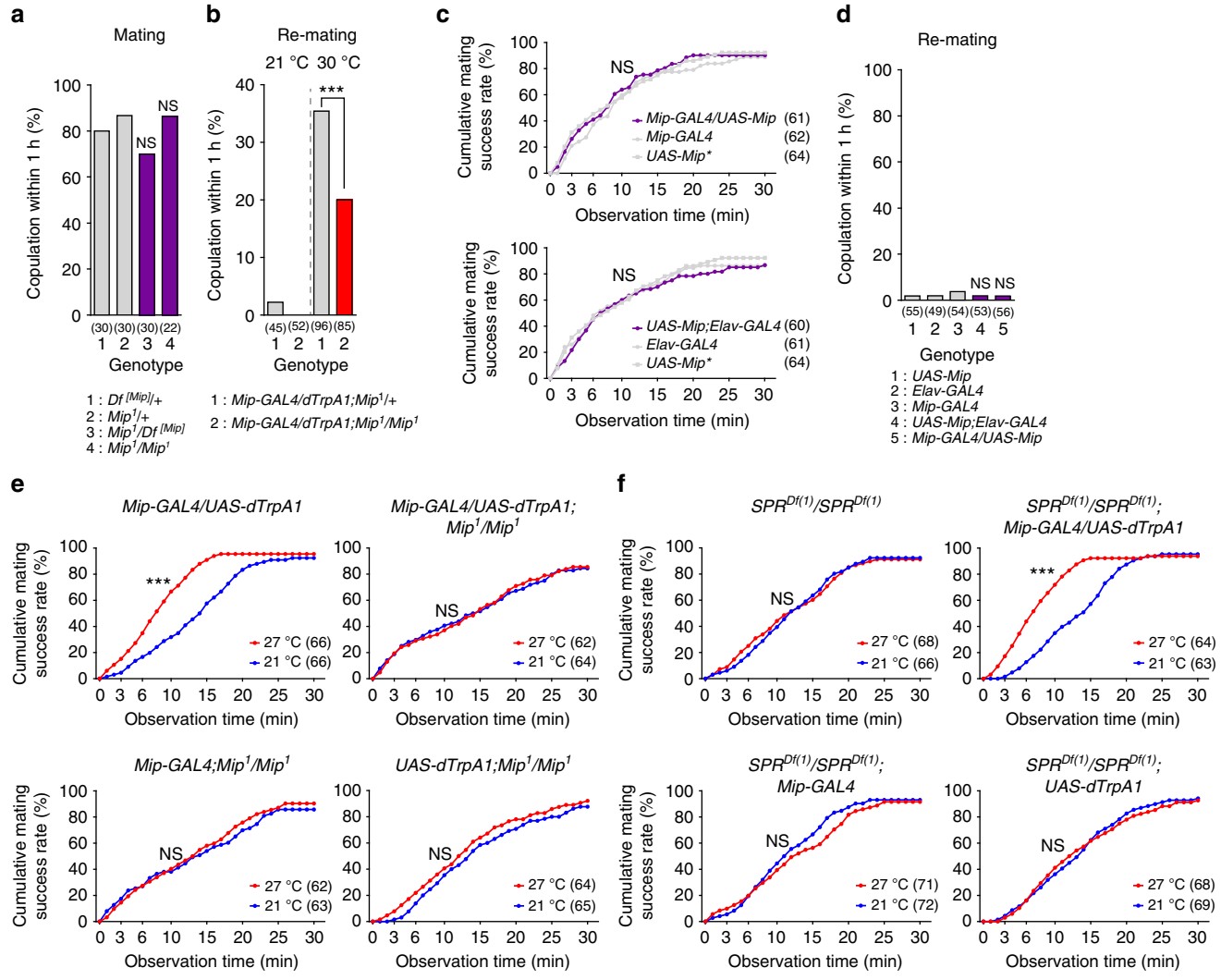

**Fig. 3** A function of Mip peptides in female mating receptivity. **a** Mating frequencies of virgin females of the indicated genotypes, scored as the percentage of females that copulate within 1 h. **b, d** Re-mating frequencies of mated females. NS indicates non-significance ($P > 0.05$); ***$P < 0.001$ for comparisons against controls; Chi-square test **a, b, d. c, e, f** Cumulative mating frequencies of virgin females. The numbers in parentheses indicate n. NS indicates non-significance ($P > 0.05$); ***$P < 0.001$ for comparisons against genetic controls (grey bars or dots in **a–d**) or a temperature control (blue in **e, f**); Chi-square test

attenuation of the TRIC signal intensity in *vAM* and SAG but not in *l-vPM* soma (Fig. 5c, d). This suggests the secretory activities of *vAM* and SAG are significantly reduced after mating. This is consistent with our behavioural observations; silencing either the *vAM* or SAG neurons induces strong mating refractoriness, a key feature of PMR. We noted that the TRIC signal changes not only in the soma but also in the neural processes (arrowheads in virgin and mated SAG in Fig. 5c). No other AG *Mip* neurons, including *l-vPM*, show mating-induced TRIC signal changes. In particular, *s-vPM* and *s-mPM* show no or very weak TRIC signal regardless of mating status (Supplementary Fig. 6). We were unable to detect any TRIC signal in *vAL* neurons, regardless of mating status (Supplementary Fig. 6). This lack of TRIC signal suggests that the activity of *vAL* neurons differs from that of *vAM* neurons.

Our finding that *vAM* activity undergoes a significant switch after mating links these neurons to the control of female mating receptivity. We noted that activation of *vAL* alone (i.e., *dsx*[FLP]-positive *Mip-GAL4* neurons) does not induce the re-mating phenotype in mated females, whereas activation of all *Mip* neurons does. Thus *vAM* neurons seem to only be partially responsible for the re-mating phenotype. To explore this idea, we adopted a mosaic approach employing an excisable GAL80[41]. We

prepared flies carrying *Mip-GAL4*, *UAS-dTrpA1*, *UAS-dsRed*, *tubulin-FRT-GAL80-FRT* ($tub > GAL80 >$) and the heatshock-flipase (*hs-FLP*). In this system, a brief exposure to high temperature (37 °C, 35 min) induces FLP expression. FLP then excises GAL80 in a small fraction of *Mip-GAL4* neurons per fly, permitting dTrpA1 and dsRed expression only in the affected cells. We proceeded to count the dsRed/dTrpA1-positive AG neurons in mosaics with and without the re-mating phenotype when they were incubated at 30 °C. Of the five groups of AG *Mip* neurons, only the *vAM* neurons show significantly more dsRed labelling in re-mating mosaics than in non-re-mating mosaics (Fig. 5e). This suggests that the sexually dimorphic *vAM* neurons are important for regulating female mating receptivity. Since we do not yet have any genetic tools for isolating *vAM* neurons at single cell resolution, we were unable to determine where and how they innervate the different parts of the CNS. A careful analysis of their TRIC labelling, however, suggests that *vAM* neurons are local interneurons with no obvious afferent processes (Supplementary Fig. 6).

**Mip neurons relay the SP signal to SAG neurons.** Our analyses suggest that the *Mip* neurons are part of the SPSN–SAG circuit

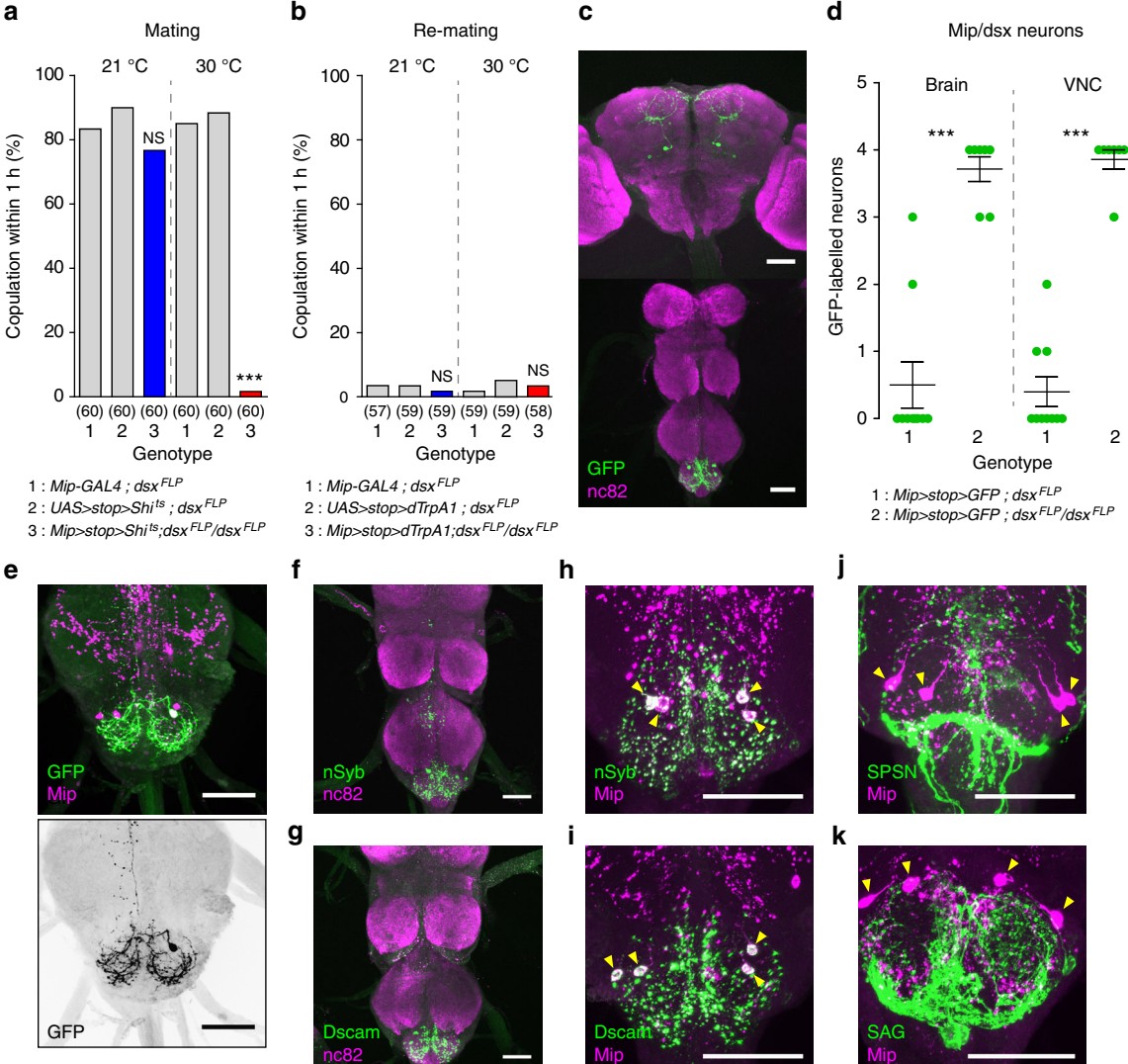

**Fig. 4** *Doublesex*-positive abdominal *Mip* neurons are important for female receptivity. **a** Mating frequencies of virgin females of the indicated genotypes, scored as the percentage of females that copulate within 1 h. **b** Re-mating frequencies of the mated females. The numbers in parentheses indicate *n*. NS indicates non-significance (*P* > 0.05); ***P* < 0.001 for comparisons against both controls (grey bars); Chi-square test **a**, **b**. **c** *Mip* and *dsx* double-positive neurons in a female carrying *Mip-GAL4, UAS > stop > mCD8-GFP* and two copies of *dsx*^FLP^ stained with anti-GFP (green) and anti-nc82 (magenta) antibodies. **d** The number of GFP-positive soma in the brain and VNC from females of the indicated genotypes. ***P* < 0.001; unpaired *t*-test. Error bars indicate s.e.m. **e** *Mip* and *dsx* double-positive *vAL* neurons in the AG of a female carrying *Mip-GAL4, UAS > stop > mCD8-GFP* and one copy of *dsx*^FLP^ stained with anti-GFP (green) and anti-Mip (magenta) antibodies (upper panel). Note that only one of the four *vAL* neurons stained in the AG of this specific preparation with the anti-Mip (magenta) antibody is GFP-positive (green). The lower panel is a negative image of the green channel from the upper panel image. Note that the single *vAL* cell extensively innervates the AG and projects one afferent axon along the midline to the thoracic ganglion. **f** *vAL* neurons from a female carrying *Mip-GAL4, UAS > stop > nSyb-GFP* and *dsx*^FLP^ stained with anti-GFP (green) and anti-nc82 (magenta). Note the localization of the presynaptic marker nSyb-GFP along the midline of the TG as well as in the AG. **g** *vAL* neurons from a female carrying *Mip-GAL4, UAS > stop > Dscam-GFP* and *dsx*^FLP^ stained with anti-GFP (green) and anti-nc82 (magenta). **h** A high magnification view of the AG from a female carrying *Mip-GAL4, UAS > stop > nSyb-GFP* and *dsx*^FLP^ stained with anti-GFP (green) and anti-Mip (magenta). Arrowheads indicate *vAL* somas **h–k**. **i** The AG from a female carrying *Mip-GAL4, UAS > stop > Dscam-GFP* and *dsx*^FLP^ stained with anti-GFP (green) and anti-Mip (magenta). **j** The AG from a female carrying *SPSN-GAL4* and *UAS-mCD8-EGFP* stained with anti-GFP (green) and anti-Mip (magenta). **k** The AG from a female carrying *SAG-GAL4* and *UAS-mCD8-EGFP* stained with anti-GFP (green) and anti-Mip (magenta). All scale bars, 50 μm

and are likely positioned between the SPSNs and the SAG neurons. This is because *Mip*-positive *vAL* neuron silencing abolishes female mating receptivity just like SAG neuron silencing (Fig. 4a) and because *vAL* processes intermingle with SPSN and SAG processes without projecting to the brain or uterus themselves (Fig. 4j, k). To test this model, we looked for functional epistasis. When we silenced either the SPSNs or SAGs with Kir2.1 (SPSN− and SAG−, respectively), we found that virgin females are

unreceptive. This is expected because SP activates an inhibitory G-protein coupled receptor (GPCR) SPR in the SPSNs, which upon activation induces PMR by silencing the SPSNs directly and SAG neurons indirectly[8, 9, 12]. We next asked whether *Mip* neuron activation can override this SPSN− condition. Indeed, *Mip* neuron activation renders virgin females receptive even in the SPSN− condition, indicating that *Mip* neurons function downstream of SPSNs (Fig. 6a). In this experiment, we observed

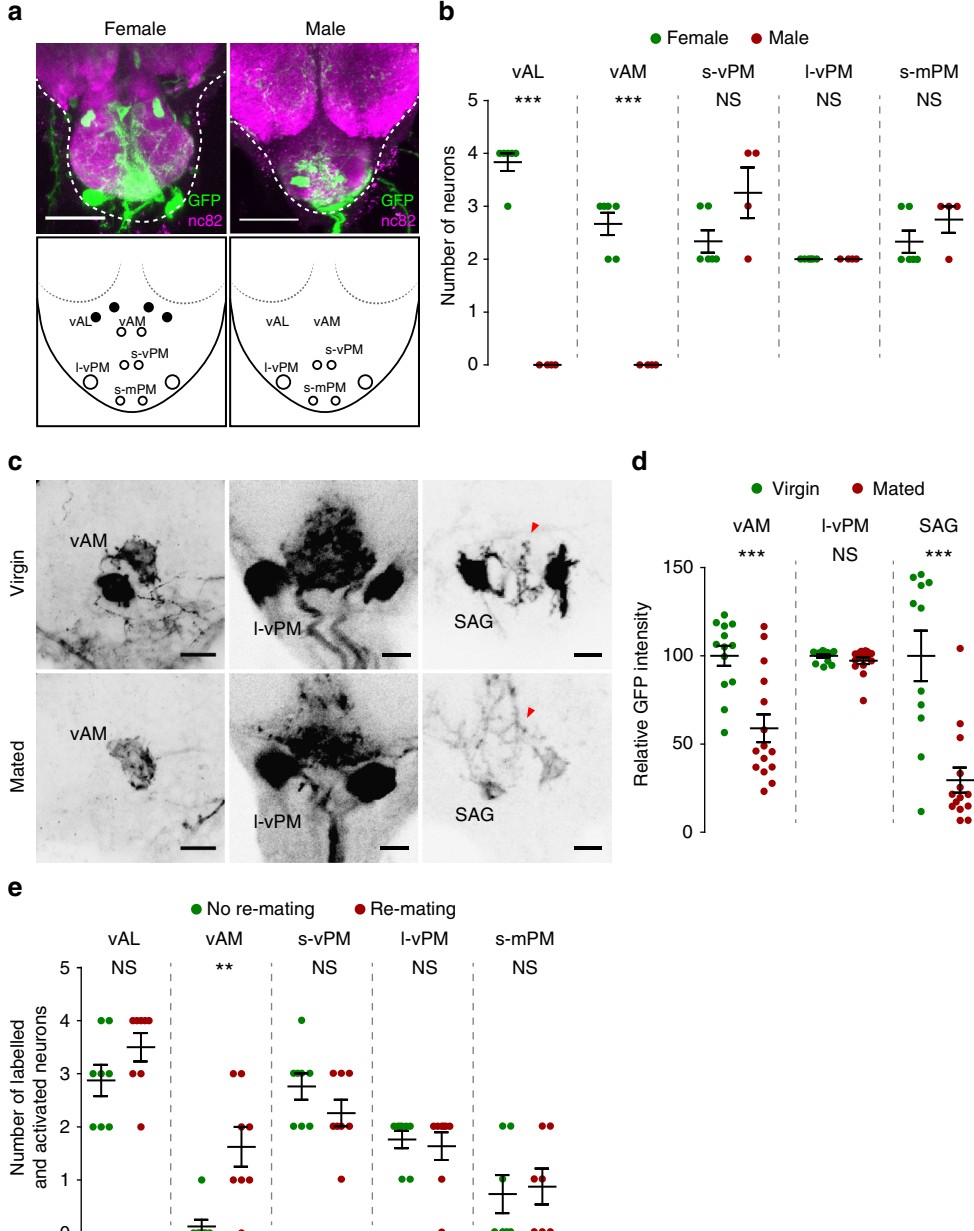

**Fig. 5** Sexually dimorphic *Mip* neurons in the AG that regulate female receptivity. **a** *Mip* neurons in the AG from a female (above left) and a male (above right) carrying *Mip-GAL4*, *Mip6.0-GAL80* and *UAS-mCD8-GFP* stained by anti-GFP (green) and anti-nc82 (magenta) antibodies. The lower panels indicate the numbers and relative locations of *Mip* neuron somas in female (left) and male (right) AGs. The *Mip* neurons in the AG are grouped into six subsets: medial anterior lateral (*mAL*), ventral anterior lateral (*vAL*), ventral anterior medial (*vAM*), small ventral posterior medial (*s-vPM*), large ventral posterior medial (*l-vPM*), and small medial posterior medial (*s-mPM*). Filled and open circles indicate neurons positive and negative for anti-Mip, repectively. Scale bars, 50 μm. **b** The numbers of GFP-positive *Mip* neurons in the AGs of females (green) and males (red) carrying *Mip-GAL4*, *Mip6.0-GAL80* and *UAS-mCD8-EGFP*. Note the clear sexual dimorphism in the *vAL* and *vAM* neurons. **c** Negative images of TRIC labelling (anti-GFP) in the AGs of virgin (upper panels) and mated females (lower panels), indicating intracellular $Ca^{2+}$ transients. Scale bars, 10 μm. **d** The GFP intensities from *vAM*, *l-vPM* and *SAG* neurons of TRIC females show $Ca^{2+}$ activity in virgin (green) and mated females (red). **e** The frequencies of labelled (and therefore activated) neurons in the indicated *Mip* neuron subset of the AG from non-re-mating (green) and re-mating mosaic females (red). We used females carrying *hsFLP*, *Mip-GAL4*, *UAS-TrpA1*, *UAS-DsRed* and *Tub FRT GAL80 FRT* for stochastic manipulation (for details, see the text). NS indicates non-significance ($P > 0.05$); \*\*$P < 0.01$, \*\*\*$P < 0.001$. Unpaired *t*-test for **b**, **d**, **e**. Error bars indicate s.e.m.

only partial restoration of mating receptivity (~ 40%) because *Mip-GAL4* targets both mating-promoting neurons (i.e., *vAL* and *vAM*) and mating-suppressing neurons (i.e., those that express *Mip^{6.0}-GAL4*, see Fig. 2d). Finally, we asked whether *Mip* neuron activation restores mating receptivity in the SAG− condition and found that it does not (Fig. 6b). This is consistent with our model that *Mip* neurons function upstream of SAGs. Together, these

observations suggest that *Mip* neurons are indeed part of the SPSN–SAG circuit and that they relay the SP signal from SPSNs to SAGs (Fig. 6c).

## Discussion

To make a mating decision, *Drosophila* females compute the sensory inputs they receive from courting males and integrate

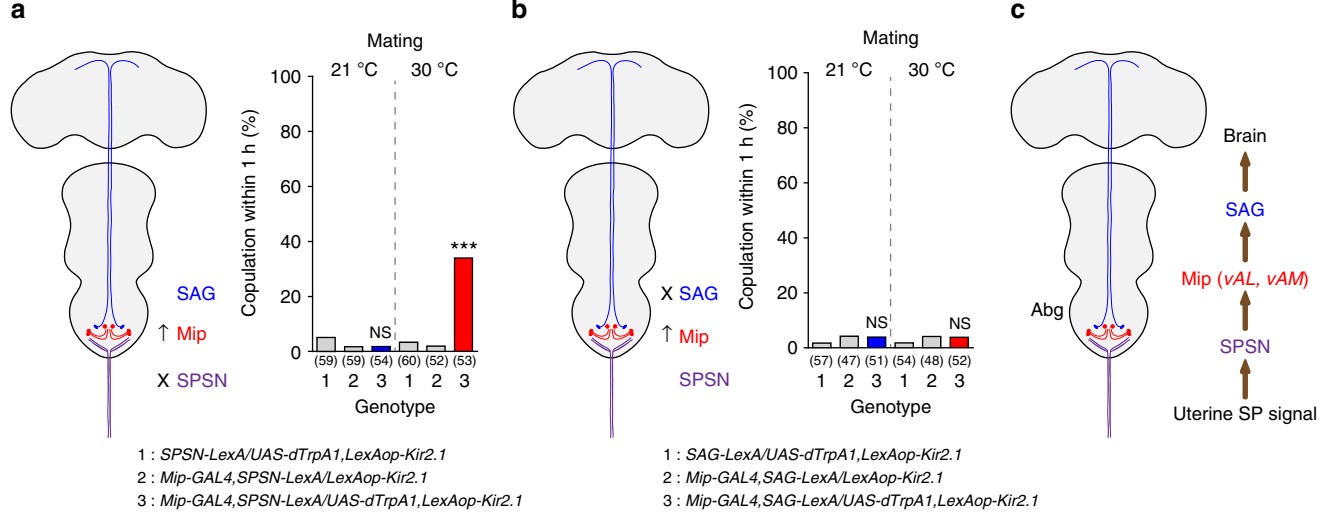

**Fig. 6** *Mip* neurons positioned within the SPSN and SAG signalling axis. **a**, **b** Mating frequencies of virgin females of the indicated genotypes, scored as the percentage of females that copulate within 1h. The numbers in parentheses indicate *n*. NS indicates non-significance ($P > 0.05$); ***$P < 0.001$ for comparisons against both controls (grey bars); Chi-square test **a**, **b**. **c** Model explaining the hierarchical relationships of the various components of the AG circuits that control female receptivity (for details, see the text)

them with the internal state coding SP signals originating in their reproductive organs. An early gynandromorph study[42] mapped this computational function to the brain. More recently, a small number of *dsx*-positive brain neurons were found to receive courtship-related olfactory and auditory inputs and to regulate female mating[43]. On the other hand, the uterine SP signal is relayed to the central brain through the SPSN–SAG circuit. By combining channelrhodopsin-mediated activation of SPSNs and patch-recording of the SAG neurons, SPSN activation was shown to elicit excitatory postsynaptic potentials (EPSPs) in SAG neurons[12]. Because the average latency from SPSN activation to the corresponding SAG EPSP is ~ 20 ms—far longer than the 1–2 ms expected for a monosynaptic connection—the SPSN–SAG connection is likely polysynaptic. Furthermore, we noted that mating attenuates not only the frequency but also the amplitude of SAG EPSPs triggered by SPSN activations[12]. This amplitude modulation seems to suggest that SAG neurons receive multiple synaptic inputs. It is possible that SPSNs modulate *vAM* and *vAL* neurons, which then feed excitatory synaptic inputs to SAG neurons (Fig. 4f). Consistent with this model, *Mip* neuron activation overrides SP injection or SPSN silencing but not SAG silencing. This functional epistasis suggests *Mip* neurons (i.e., *vAL* and *vAM*) function downstream of SPSNs and upstream of SAGs. Because SP can access central neurons via the haemolymph, SP may modulate SAG EPSPs by directly acting on the *vAM* and *vAL* neurons. Since *Mip* neuron-specific mSP expression elicits no sign of PMR-like behaviours, SP actions on those are less likely.

Our genetic analyses of the *Mip* mutants indicate that the Mip neuropeptides can promote mating receptivity both in virgin and mated females. Mip is a potent and physiologically relevant agonist for SPR[32, 33]. Like other neuromodulators, Mip is highly pleiotropic, being involved in behaviours as diverse as sleep and feeding[26, 44]. SPR mediates the sleep function of Mip but is dispensable for its feeding function[26, 44]. Our observation that the mating function of Mip does not require SPR suggests the presence of another unknown Mip receptor (hereafter referred to as hypothetical Mip receptor 1 or hMipR1).

The SPSN–SAG circuit fires more frequently in virgin females than in mated females. Since *vAL* neurons are a part of the SPSN–SAG circuit, they should actively release Mip neuropeptides in virgin females. These, in turn, should activate SPR at the

SPSN axon terminals and induce PMR. We found, however, that rather than supressing mating receptivity, hyper-activation of *vAL* increases it. Why is this? The simplest explanation is that SPR is absent from the SPSN axon terminals exposed to Mip. The subcellular localization of most metabotropic neurotransmitter receptors and other GPCRs is not tightly controlled and many of these proteins show extrasynaptic localization[45]. Alternatively, if Mip activates both SPR and hMipR1 and if they suppress and promote mating receptivity, respectively, the activation of hMipR1 may dominate the effect of SPR activation. SPR deficiency, however, has little impact on the mating induced by *Mip* neuron activation. This suggests that SPR activation by Mip does not induce PMR-like mating refractoriness. It also suggests that Mip and SP are biased GPCR ligands that recruit different signalling pathways downstream of SPR[46]. We, therefore, propose the following scenario: Activation of *vAL* neurons produces virgin-like mating receptivity because Mip released from *vAL* neurons activates hMipR1 in SAG neurons while simultaneously desensitizing SPR locally in SPSN processes that arborize in the AG. *vAM* neurons also contribute to this process by releasing unknown substances that synergize with the actions of Mip (Fig. 6c).

## Methods

**Fly stocks**. Fly lines were raised at 25 °C and 60% humidity in 12 h:12 h light:dark cycle with the standard fly media. Following stocks are reported previously or obtained from the Bloomington Drosophila Stock Center (BDSC): *Ccap-GAL4*[47], *AstCC-GAL4*[48], *Pburs-GAL4*[48], *Mip6.0-GAL80*[48], *Kinin (drosokinin)-GAL4*[26], *Mip-GAL4*[26], *Mip1*[26], *UAS-Mip*[26], *UAS-dTrpA1*[25], *UAS-Shi*[24], *UAS-SPR-IR*[8], *UAS-SPR*[8], *UAS > Shi*[ts49], *UAS > stop > dTrpA1*[50], *fru*[FLP51], *UAS > stop > mCD8GFP*[51], *UAS > stop > nSybGFP*[51], *UAS > stop > Dscam17.1-GFP*[51], *dsx*[FLP21], *UAS > stop > Kir2.1* (a gift from Barry J. Dickson, Janelia Research Campus), *LexAOP-Kir2.1*[12], *SPSN-GAL4 (VT3280-GAL4)*[12], *SPSN-LexA (VT3280-LexA)*[12], *SAG-GAL4 (VT50405-p65AD; VT7068-GAL4DBD)*[12], *SAG-LexA (VT50405-LexA)*[12], *UAS-mSP*[30], *otd*[FLP39], *ppk-GAL4*[10], *UAS-mCD8-GFP* (a gift from Barry J. Dickson, Janelia Research Campus), *189y-GAL4* (BDSC stock number, 30817), *36y-GAL4* (30819), *Ddc-GAL4* (7009), *Trh-GAL4* (38388), *Ple-GAL4* (8848), *UAS-Dicer2* (24648), *Df(3 L)Exel6131* (7610), *Elav-GAL4* (8760), *Df(1)Exel6234* (7708), *UAS-mCD8::RFP*, *LexAop2-mCD8::GFP;nSyb-MKII::nlsLexADBDo;UAS-p65AD:: CaM* (61679), *hsFLP122* (1929), and *Tub > GAL80 >* (38879). Other stocks are generated in this study (see below).

**Molecular biology**. *Amn-GAL4*, *CCHa1-GAL4*, *dilp7-GAL4*, *Dms-GAL4*, *Nplp2-GAL4* and *Nplp4-GAL4* were prepared in pAGAL4, and *Mip6.0-GAL4*, *MipA-GAL4*,

$Mip^B$-GAL4, $Mip^C$-GAL4 and $Mip^D$-GAL4 were prepared in the gateway vectors as described previously[52]. The GAL4 transgene was inserted into a specific site of second chromosome (VIE-72A, a gift from Barry J. Dickson, Janelia Research Campus) using ΦC31 system[53]. pAGAL4 is prepared by inserting a site-specific integration site (attB) into 7–74 site of pPTGAL4(+) vector[54]. Genomic fragments and primer sequences used to generate these are as follows: Amn-GAL4 (−3493 to +132; tatagcggccgctttcggtgggaagttagtgc, gcgcgctctagatgtacatataggcccgtcgtc), CCHa1-GAL4 (−619 to +180; tatagcggccgcttctgcctgctatgacggttg, gcgcgctcta-gacgctctacctcaacacggtct), dilp7-GAL4 (−1181 to +35; tatagcggccgcgccaggcaaa-taaattcagc, gcgcgctctagacagctgcccgagtttttgtat), Dms-GAL4 (−2044 to +96; tatagcggccgctcttcctcactgcattagtcacg, gcgcgctctagagcatagaggtggaccctgaa), Nplp2-GAL4 (−1446 to +12; tatagcggccgccagcatcgccacttcacattt, gcgcgctctagagagctggccatttttgtgt), Nplp4-GAL4 (−832 to +149; tatagcggccgcgccggaatagaagtcgatga, gcgcgctcta-gaaaccagctgggaaaggaaaa), $Mip^A$-GAL4 (−5928 to −4626; caccagcagcaaaaagtcggaaaa, gagttcgtccatccaggaga), $Mip^B$-GAL4 (−4626 to −3136; cacctctcctggatggacgaactc, gaaaaggctggctttgtctg), $Mip^C$-GAL4 (−3134 to −1468; caccgacaaagccagccttttcaa, cccaaaacttgcctacaacc), and $Mip^D$-GAL4 (-1521 to +51; caccgggatggttgtaggcaagtt, gaggagcaccatcagaaagc).

**Behaviour assays.** For behaviour assays, we followed procedures described previously[8]. Virgin males and females were collected at eclosion. Males were aged individually for 5 days; females were aged for 3 days in groups of 15–18. All assays were performed at zeitgeber time (ZT) 6:00–12:00, and repeated on at least two different days. For the mating assay, single virgin females and naive CS males were paired in 10-mm diameter chambers and videotaped (SONY DCR-SR47) for 1 h. The females that copulated were then transferred individually to food vials for 48 h. For the re-mating assay, the mated females were re-tested in the same manner with naive CS males. For rejection behaviours, either single virgin or mated females were paired with naive CS males, videotaped for 10 min at higher magnification and scored manually for ovipositor extrusions. Egg laying assays were performed in vials containing 1.5 ml of standard fly media. Virgin females were transferred to vials in groups of five for the virgin egg laying assay, and mated females were individually moved to vials for the mated egg laying assays. They were allowed to lay eggs for 48 h. The eggs laid from five virgins or one copulated female were manually counted using a stereeromicroscope (ZEISS Stemi DV4).

**Immunohistochemistry.** Unless stated otherwise, 3–5-day-old virgin female or male flies were dissected under phosphate-buffered saline (PBS; pH 7.4). Tissues were fixed for 30 min at room temperature in 4% paraformaldehyde in PBS. After extensive washing, the tissues were incubated in primary antibody for 48 h at 4 °C and in secondary antibody for 24 h at 4 °C. Antibodies used were: rabbit anti-GFP (1:1000; Invitrogen, A11122), mouse anti-GFP (1:1000; Sigma, G6539), rabbit anti-DsRed (1:1000; Clontech, 632496), mouse anti-nc82 (1:50; Developmental Studies Hybridoma Bank), Alexa 488-conjugated goat anti-mouse (1:1000; Invitrogen, A11001), Alexa 488-conjugated goat anti-rabbit (1:1000; Invitrogen, A11008), Alexa 568-conjugated goat anti-mouse (1:1000; Invitrogen, A11004), and Alexa 568-conjugated goat anti-rabbit (1:1000; Invitrogen, A11011). The CNS was mounted in Vectashield (Vector Laboratory, H-1000). Images were acquired with Zeiss LSM 700/Axiovert 200 M (Zeiss), and were processed in Image J[55].

**TRIC analysis.** Mip-GAL4 or SAG-GAL4 (VT50405-p65AD; VT7068-GAL4DBD) flies were crossed with a TRIC line (BDSC stock number, 61679; see above) to quantify the postmating changes in intracellular $Ca^{2+}$ levels in vAM and l-vPM or SAG neurons, respectively. CNS tissues from 5-day-old virgin or mated females (2 days after copulation) were processed for anti-EGFP staining as described above (see Immunohistochemistry section). To quantify EGFP fluorescence, maximum intensity Z-projections of three consecutive confocal stacks (3.82 μm-thick each) covering the entire soma of each neuron were merged with Image J. Then the relative GFP intensity of each soma was calculated by setting the average of the somas in each group from virgin females as 100%.

**Mosaic analysis.** For the mosaic analysis, we followed the procedure described in Gordon and Scott[41] with minor modifications. To generate small numbers of GAL80-negative cells, flies of the genotype tub > GAL80 > ;Mip-GAL4/UAS-dTrpA1;hs-FLP/UAS-dsRed were raised at 23 °C and subjected to a brief (~ 35 min) heat shock at 37 °C. The mosaic females were collected shortly after eclosion and aged for 3 days in groups of ~ 10. Each virgin female was then mated with naive CS males individually at 23 °C, and the mated females were kept in a single vial for 2 days. For the second mating assays, the mated mosaic females were paired with naive CS males at 30 °C for 1 h. Mosaics were separated into two groups: re-mating and non-re-mating. Induction of FLP expression earlier in development resulted in more mosaic flies with the re-mating phenotype, yet many more neurons expressing dsRed. To achieve maximally restricted labelling, FLP expression was induced at the late pupal stage (10 days after egg laying). With this protocol, the re-mating phenotype was observed in approximately 20–30% of mosaics, approximately ~ 25% more than those without FLP induction. To assess identities of activated neurons (presumably expressing both dTrpA1 and DsRed), the CNS was dissected, fixed and mounted for inspection of DsRed-expressing neurons as described in the Immunohistochemistry section.

**Statistical analysis.** Statistical analyses were performed using Prism (GraphPad Software).

**Data availability.** The authors declare that all data supporting the findings of this study are available within the paper and its Supplementary Information files.

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

## Acknowledgements

We thank H-S. Yoon, J. Mun and J. Song for excellent technical assistance and W.D. Jones (Korean Advanced Institute of Science and Technology) and D. Yamamoto (Tohoku University) for helpful advice on the manuscript. This work was supported by the GIST Research Institute (GRI) grant funded by the GIST in 2017 and by Basic Science Research Programs (NRF2013-R1A1A2010475, NRF-2015R1A2A1A10054304, NRF-2015K2A1B8046794, NRF-2017M3A9B8069650) through the National Research Foundation of Korea (NRF) funded by Ministry of Science, ICT and Future Planning (MSIP), Republic of Korea. Stocks obtained from the Bloomington Drosophila Stock Center (NIH P40OD018537) and the Korea Drosophila Resource Center were used.

## Author contributions

Y.-H.J. and H.-S.C. performed experiments. Y.-J.K. supervised the project and wrote the manuscript.

## Additional information

**Competing interests:** The authors declare no competing financial interests.

