## [Peer Review File · Nature Communications]

Reviewers' comments:

Reviewer #1 (Remarks to the Author):

These authors investigated to find interneuronal microcircuits that are essential to modulate female fly's mating receptivity. To identify the subpopulation of interneurons, authors screened previously defined neuropeptide GAL4s by combining them with UAS-dTrpA1 or UAS-shits to activate or silence GAL4-labeled neurons. In the activation screening, authors found that mated females frequently remate after 48 hours when MIP-GAL4 labelled cells are activated by UAS-dTrpA1. By the series of experiments in Figure 1, authors identified that the neurons labelled by Mip-GAL4 are composed of mixed populations of interneurons. While narrowing down the circuits, they identified that Mip6.0-GAL80 labelled cells are responsible for remating phenotype induced by UAS-dTrpA1. Sex peptide (SP) and its receptor SPR signalling pathway have been known to be critical components of female receptivity and post-mating responses. By the series of genetic dissections, authors successfully identified that the remating phenotype induced by Mip-GAL4>UAS-dTrpA1 is downstream of SP pathway. They also identified that MIP itself is important for signalling inside the circuits, however not through its potent agonist SPR. Authors suggest that there may be hypothetical MIP receptor that is responsible for mediating the receptivity signalling through neuropeptide MIP. Next, authors performed intersectional methods using fruFLP, dsxFLP, and otdFLP combined stop cassette containing UAS transgenes. In these efforts, they identified that there are female-specific doublesex-positive interneuronal populations in the VNC. By using transcriptional calcium reporter and mosaic analysis, authors successfully identified that vAM neurons are the possible candidate interneurons bridge the SP signal from SPSN to SAG. In conclusion, these authors identified secondary ordered neurons that are vital to modulate female's receptivity. Further analysis of their screening data will help to understand how females can make mating decisions dependent upon their internal mating status.

1. Authors performed well-controlled genetic experiments to identify the interneuronal microcircuits that are difficult to dissect when they are associated with complex behavioural traits. As authors already pointed out, the SPSN is the primary sensory neurons located in the female uterus. Authors need to show that SPSN postsynaptic terminals are connected to vAL or vAM. This can be done simply by using ppk-CD4-tdGFP lines (Bloomington 35843) crossed to Mip-GAL4>UAS-tdTomato flies. If these flies can be combined to fru-GAL80, they can confirm that SPSN synaptically connected to vAL or vAM.

2. Authors did not show the imaging data of Mip-GAL4 combined to Mip-GAL80 they generated. Since they already performed the functional test with Mip-GAL80, it will be interesting which interneuronal populations are labelled by Mip-GAL4/Mip-GAL80 combination and whether it can be linked to the functional data they generated in Fig. 1d-j.

3. Since these neurons show the female-specific phenotype, it will be interesting what will happen if these neurons become masculinized. Authors simply can perform experiments using UAS-dsxM crossed to Mip-GAL4 then test the remating phenotype resulting from them.

All the works authors performed are impressive and promising results that will help the Drosophila research community to increase the knowledge of how the neuropeptide signalling can slowly and reversibly modulate interneuronal network and results in significantly altered behaviour. I strongly suggest that Editor should publish this manuscript in Nature Communications.

Reviewer #2 (Remarks to the Author):

Jang and colleagues investigate the neural circuit underpinning Drosophila melanogaster female receptivity to re-mating. They focus on neurons that express the neuropeptide Mip based on results of a screen for neuromodulators involved in this behavioral switch. The question is interesting, the approach is very good, the results are of high quality. Using a combination of genetic tools including a Mip-Gal4 and a Mip-Gal80 line they genetically dissect out two sets of neurons in the fly's VNC. Based on another set of results (with dsx-flp and otd-flp) the authors propose that in particular the so-called vAL neurons promote female mating receptivity.

While this manuscript clearly has the potential to extend our knowledge of the circuit that controls female receptivity, it, unfortunately, also leaves several open questions and knots untightened. The model (Figure 4f) therefore appears quite vague, because the functional connectivity of the neurons is still somewhat elusive. This carries over into the discussion, which is brief and to the point, but again shows that many issues remain unresolved, including the role of SPR, the relative role of vAL and mAL, the role of Mip itself in these different neuron populations and its relationship to SP. It is clear that the authors will not be able to solve the puzzle completely, but at this point, this reviewer feels left with more questions than answers. I suggest to either solidify the role of SPR or remove this part altogether, because the results are not fully convincing (to their credit the authors point this out themselves in the discussion). Furthermore, providing anatomical evidence for the presence of the Mip subpopulations, the effect of Gal80, the effect of one versus two copies of dsx-flp and perhaps a clearer indication for the reader, which neuron subsets are activated or silenced in the different experiments (or even in the figures) could make it easier to follow the logic of the paper.

Technically, the paper is of high quality with careful behavioral analysis including the statistics, beautiful images and state-of-the-art methods.

Additional comments:

1. Page numbers are missing, which is not helpful.
2. P3: This is a strange thing to say: 'Our analysis of the Dh44-GAL4 results will be published elsewhere'. I suggest to remove it.
3. P4: clearly state the molecular difference between the driver in Mip-Gal4 and Mip-Gal80 in the main text. In addition, images comparing GFP expression with and without Gal80

should be provided to help understand the behavioural phenotype.

4. P4: given the importance of the ovipositor extrusion result, I suggest to include representative videos showing the respective phenotypes.

5. P4: given the phenotype of ovipositor extrusion, I find it problematic to talk about a 'mating decision' in the context of this assay. Do the females 'decide' not to remate or the males cannot mate with them, because they extrude their ovipositor. This should be rephrased to allow for different interpretations.

6. P4: 'Because there were no Mip-GAL4-positive ppk neurons in the uterus (not shown)...'. This result should be shown given its importance.

7. P5: 're-mating frequency in mated females (Fig. 2c)'... The label above the figure panel states 'mating' and not 're-mating'. This should be corrected.

8. P6: 'We used 27 °C for thermal activation to minimize the activation of the subpopulation of Mip neurons that inhibits mating (i.e., those that express Mip6.0-GAL80, see Fig. 1).' I do not understand the rationale here. Please explain. Why does 27 degrees affect these neurons differentially as compared to the non-Gal80 populations?

9. Figure 2e: the result that Mip neuron activation enhances mating in the absence of SPR is very interesting. Depending on the effect of Mip on SPR signalling (activating or inhibiting (possibly depending on Mip levels or cell state), however, this result might be expected although SPR is the receptor for Mip. I suggest to include SPR/SPR mutants in the absence of Mip-TrpA1 and test them at 31 degrees to exclude that the effect of temperature in the background of SPR mutants contributes or explains this effect.

10. Figure 3b: Why was the re-mating assay not carried out for 2 copy dsx-flp animals? Otd-flp only addresses the brain neurons, but not the additional neurons labelled in the VNC...

11. P8: The authors state: 'We noted that activation of vAL alone does not induce the re-mating phenotype in mated females, whereas activation of all Mip neurons does.' I assume they refer to the dsx-flp results. If so, this should be made clear. If not, they should name the evidence.

Reviewer #3 (Remarks to the Author):

This is an interesting study that uses manipulation of specific peptide-containing neurons in *Drosophila* to examine the neural circuitry underlying female sexual receptivity. This field has produced many novel findings and the authors have previously contributed important findings, especially regarding the nature of the peptide ligands and receptors involved. I have several questions and comments that may help improve this manuscript:

Lines 23-25

The summary finishes with the sentence:

"Genetic analyses with a Mip-null mutant suggest that the Mip neuropeptide produced in vAL promotes mating receptivity both in virgins and mated females.

But lines 144-145 states that MIP -/- virgins mate normally – so I do not understand how this conclusion (the main one) is reached.

Line 73

"Silencing of EH, capa and kinin gal4 neurons produces a marginal or non-significant phenotype..." From the Methods, I understand "marginal" to mean $p < 0.05$ or $p < 0.001$ – that sounds significant by every conventional statistical approach I am used to....so what is meant by the term 'marginal'? non-significant? The authors should explain their thinking on this.

S Figure 1

Not sure why MIP-Gal4>UAS-shi (21 and 30 degrees) in mated females is not reported - please include or explain.

Line 95

The 6 kB gal80 line appears to be a very precise discriminator for presumed 'inhibitory' MIP neurons. Was this fragment chosen arbitrarily? Were any others tried? It would be useful to see a schematic map of MIPGAL4+ neurons and MIP-Gal80[6.0]+ neurons. I would have more confidence in the interpretation if the authors had explored the result more with additional constructs/experiments. Were all inhibitory neurons removed? Were any excitatory neurons silenced?

Lines 113-114

Please provide a reference for statement regarding egg-laying.

Line 135

The use of tethered SP is useful. Have the authors tried a tethered MIP? That would seem to be a useful reagent in this analysis. What evidence exists to indicate the transmitter phenotype of the SPSN neurons that project from the uterus?

Line 152

The text does not reference the correct panels of Figure 2. Not sure the significance of Figure 2C, as MIP cell activation is supposed to increase receptivity, but in virgin females, it is already near 100% - therefore this experiment appears hampered by a ceiling effect.

Line 162

Unexpectedly, the design switches protocols for thermal activation: previously it used 30 degrees (e.g. Figure 1), now it switches to 27 degrees. The text explains that use of 27 degrees was intended to "minimize the activation" of the inhibitory MIP cells – on what basis was this predicted? How do the authors know the inhibitory neurons require a higher thermal threshold for activation – is there some evidence? Without some plausible

explanation, I find this set of experiments very difficult to evaluate.

Figure 2

In Figure 2b, why not add the Mip-Gal80 to test the contribution of the MIP gene to the "full" complement of activating MIP+ neurons?

In Figure 2d, mipGal4>TrpA1 at 27 degrees produces 100% cumulative mating in less than 20 min. However, that genotype at 30 degrees produces ~60% copulation within 1 hr. Aren't these the same behavioral measure, and if so why are the outcomes discrepant? If different behavioral measures, please explain the difference in the assay, and also why two different assays were used.

Line 192

Silencing Mip/dsx neurons decreased virgin female receptivity (Figure 3a); yet the text states these neurons "maintains female sexual receptivity". I disagree – at most, the evidence supports the hypothesis that they elevate virgin female receptivity. BTW, 3a uses Kir to silence neurons whereas other experiments use UAS-shi: any explanation for using different reagents to the same end? This adds design variables that lessen the strength of the conclusion regarding neuronal function.

Lines 201-210

The section using confocal microscopy to interpret synaptic interconnections between SPSN and vAL and SAG neurons is not credible. Apparent proximity at the LM level interpreted to represent "reciprocal connections" is not a conservative interpretation.

Lines 211-250

I was confused by this section in that the text no longer referred to dsx neurons – why not?

Line 215

EGFP labeled Gal4 neurons are not specified as to age or mating status.

Figure 4a

I do not understand the female/male comparison in this panel. The cartoon does not correspond to the image shown. The male neurons indicated by a star appear by position, number, size and shape to correspond to the vAM of females, but instead are labeled s-vPM. The authors need to clarify their interpretation and present criteria by which they assigned these cell identities. Also what is the point of the star next to the three midline neurons in the male example?

Figure 4c

The low power images are of little value without some description. A high power view is offered of the vAM region (I cannot really see the yellow dashed box): How were these two examples (the ones illustrated for us) scored quantitatively in panel e? I think I see 2 cells in the virgin and 2 cells in the mated - is that what the authors scored? In the lateral aspects of T3, the virgin tissue contains 1-2 dark cells and the mated tissue appears to contain these same cells, but the latter are much less intensely stained. That raises the question - is diminished TRIC staining exclusively displayed by vAM neurons? If not, does

diminished staining in the vAM retain as much significance? What are the criteria employed here to score a positive cell body? Were all labeled cell groups scored? Is it fair to only compare cell body staining with this method? There appears to be considerable staining of processes. Finally, can the authors demonstrate the validity of the assay (the staining and scoring methods) with some positive control set of neurons, for which there is independent evidence of activity in a given interval? i.e., a positive control.

Line 245

Typo: 4g should be 4e.

> We performed additional experiments to examine the functional relationships between the *Mip* neurons (*vAL/vAM*), SPSNs, and SAG (See new Fig. 6). Our functional epistasis experiments show that *Mip* neuron activations can override SPSN silencing, but not SAG silencing. This indicates the *Mip* neurons are positioned between SPSNs and SAG neurons in this neuronal circuit. This observation, together with other anatomical and behavioral data suggest that *Mip* neurons likely relay signals from SPSNs to SAGs, which project to the brain.

In response to the second comment on characterizing the *Mip-GAL4* and *Mip6.0-GAL80* subsets, we prepared a new Fig. 2 that shows *Mip-GAL4*, *Mip6.0-GAL4* and *Mip6.0-GAL80* expression in the CNS stained with anti-MIP. Schematic diagrams are also included in Fig. 2. For more details, please see our response to comment 2 of Reviewer 1 below.

Reviewers' comments:

Reviewer #1 (Remarks to the Author):

1. Authors performed well-controlled genetic experiments to identify the interneuronal microcircuits that are difficult to dissect when they are associated with complex behavioural traits. As authors already pointed out, the SPSN is the primary sensory neurons located in the female uterus. Authors need to show that SPSN postsynaptic terminals are connected to vAL or vAM. This can be done simply by using ppk-CD4-tdGFP lines (Bloomington 35843) crossed to Mip-GAL4>UAS-tdTomato flies. If these flies can be combined to fru-GAL80, they can confirm that SPSN synaptically connected to vAL or vAM.

> Indeed, we performed a GFP Reconstitution Across Synaptic Partners (GRASP) experiment to visualize potential synaptic contacts between *Mip-GAL4* neurons (including *vAL* and *vAM*) and *ppk* neurons (see Figure P1 below). The GRASP results, however, were difficult to interpret because the *ppk-LexA* and *Mip-GAL4* expression were not limited to the processes of the SPSNs and *vAL* (or *vAM*), respectively. *Mip-GAL4* expression occurs not only in *vAL* and *vAM*, but also several additional AG neurons (Fig. P1A). Likewise, *ppk-LexA* is expressed in SPSNs and many additional *ppk*-positive sensory neurons, many of the processes of which enter the AG (Fig. P1B). Although we detected robust GRASP signals between *ppk-LexA* and *Mip-GAL4* processes (Fig. P1C), we were unable to distinguish the GRASP signal coming from the contact points between *vAL* (or *vAM*) and SPSNs and that coming from other functionally irrelevant *ppk-LexA* and *Mip-GAL4* contacts. We, therefore, are unable to offer new data in response to this point because of a technical reason.

Figure P1 | Cellular contacts between *Mip* and *ppk* neurons in the abdominal ganglion.

(A, A') Confocal sections of the posterior VNC (TG3 and AG) displaying GFP fluorescence from a *Mip-Gal4 Mip6.0-Gal80 UAS-mCD8-EGFP* female. Ventral (A) and lateral views (A') are presented. Yellow arrowheads and arrows indicate *vAL* and *vAM* neurons, respectively. White arrows indicate the neurons labeled in a control lacking *ppk-LexA::VP16* (white arrows in D).

(B, B') Confocal sections of the posterior VNC displaying GFP fluorescence from a *ppk-LexA LexAOP-mCD8-EGFP* female. Ventral (B) and lateral views (B') are presented.

(C, C') Confocal sections of the posterior VNC displaying intensity of GRASP labeling in a red-hot color scale, from a *ppk-LexA LexAop-CD4::spGFP11 Mip-Gal4 UAS-CD4::spGFP1-10* female. Arrowheads (white) indicate the neurons labeled in a control lacking *ppk-LexA::VP16* (white arrows in D).

(D, D') Confocal sections of the posterior VNC displaying GRASP labeling from a control female lacking *ppk-LexA::VP16* (*LexAop-CD4::spGFP11/Mip-Gal4;UAS-CD4::spGFP1-10/+*). Note that one pair of neurons at the tip of the abdominal ganglion (white arrows) shows strong GRASP signal even without *ppk-LexA::VP16*. The signal from these neurons is restricted to the distal tip of the AG. TG, thoracic ganglion; AG, abdominal ganglion. Scale bars, 50 μ m.

2. Authors did not show the imaging data of *Mip-GAL4* combined to *Mip-GAL80* they generated. Since they already performed the functional test with *Mip-GAL80*, it will be interesting which interneuronal populations are labelled by *Mip-GAL4/Mip-GAL80* combination and whether it can be linked to the functional data they generated in Fig. 1d-j.

> We carefully mapped the CNS neurons expressing *Mip-GAL4*, *Mip6.0-GAL4* or *Mip6.0-GAL80* in females. To compare expression of these transgenes, we combined *GAL4* or *GAL4/Gal80* with *UAS-mCD8-EGFP*, and stained the CNS expressing EGFP with anti-EGFP and anti-Mip. Using anti-Mip staining as a reference, we compared *Mip*⁺>EGFP, *Mip6.0*⁺>EGFP, and *Mip*⁺,*Mip6.0*⁺>EGFP. Schematics indicating the positions of anti-Mip and/or anti-EGFP positive neurons are shown (see Fig. 2a-c).

[redacted]

All the works authors performed are impressive and promising results that will help the Drosophila research community to increase the knowledge of how the neuropeptide signalling can slowly and reversibly modulate interneuronal network and results in significantly altered behaviour. I strongly suggest that Editor should publish this manuscript in Nature Communications.

Reviewer #2 (Remarks to the Author):

*Jang and colleagues investigate the neural circuit underpinning *Drosophila melanogaster* female receptivity to re-mating. They focus on neurons that express the neuropeptide *Mip* based on results of a screen for neuromodulators involved in this behavioral switch. The question is interesting, the approach is very good, the results are of high quality. Using a combination of genetic tools including a *Mip-Gal4* and a *Mip-Gal80* line they genetically dissect out two sets of neurons in the fly's VNC. Based on another set of results (with *dsx-flp* and *otd-flp*) the authors propose that in particular the so-called vAL neurons promote female mating receptivity. While this manuscript clearly has the potential to extend our knowledge of the circuit that controls female receptivity, it, unfortunately, also leaves several open questions and knots untightened. **The model (Figure 4f) therefore appears quite vague, because the functional connectivity of the neurons is still somewhat elusive.** This carries over into the discussion, which is brief and to the point, but again shows that many issues remain unresolved, including the role of SPR, the relative role of vAL and mAL, the role of *Mip* itself in these different neuron populations and its relationship to SP. It is clear that the authors will not be able to solve the puzzle completely, but at this point, this reviewer feels left with more questions than answers. I suggest to either solidify the role of SPR or remove this part altogether, because the results are not fully convincing (to their credit the authors point this out themselves in the discussion).*

> In this revision, we show the functional relationships among the *Mip* neurons (vAL/vAM), SPSNs, and SAG neurons (See new Fig. 6). Using functional epistasis experiments, we found that *Mip* neurons function downstream of SPSNs, and upstream of SAGs. This supports the model that *Mip* neurons relay SPSN-born SP signal to SAGs. In addition, we removed our speculation on the possibility that *Mip* competes against SP from the haemolymph for SPR (lines 283-289 in the first submission), because *Mip* activation is capable of promoting mating even when the SPSNs have been silenced. This means *Mip*'s action on the SPSNs appears to be negligible.

*Furthermore, providing anatomical evidence for the presence of the *Mip* subpopulations, the effect of *Gal80*, the effect of one versus two copies of *dsx-flp* and perhaps a clearer indication for the reader, which neuron subsets are activated or silenced in the different experiments (or even in the figures) could make it easier to follow the logic of the paper.*

> See the response to comment 2 of Reviewer 1 and the new Fig. 2a-c.

Technically, the paper is of high quality with careful behavioral analysis including the statistics, beautiful images and state-of-the-art methods.

Additional comments:

1. Page numbers are missing, which is not helpful.

> Corrected.

2. P3: *This is a strange thing to say: ‘Our analysis of the Dh44-GAL4 results will be published elsewhere’. I suggest to remove it.*

> Removed.

3. P4: *clearly state the molecular difference between the driver in Mip-Gal4 and Mip-Gal80 in the main text.*

> Corrected. See lines 93-102.

In addition, images comparing GFP expression with and without Gal80 should be provided to help understand the behavioural phenotype.

> See the response to comment 2 of reviewer 1.

4. P4: *given the importance of the ovipositor extrusion result, I suggest to include representative videos showing the respective phenotypes.*

> Included as Supplemental video 1.

5. P4: *given the phenotype of ovipositor extrusion, I find it problematic to talk about a ‘mating decision’ in the context of this assay. Do the females ‘decide’ not to remate or the males cannot mate with them, because they extrude their ovipositor. This should be rephrased to allow for different interpretations.*

> We revised the text to remove the term ‘mating decision.’

> In our first submission, we did not exclude the possibility that *Mip* neurons are simply involved in motor activities generating ovipositor extrusion behaviour. In this revision, we now show virgin females with silenced *Mip* neurons show rejection behavior only when they are courted (Fig. 11). We showed this in two ways. First, we showed that *Mip* neuron silencing itself does not induce ovipositor extrusion in isolated virgin females. Next, we repeated the same result in females kept with non-courting males, such as *fru^F* males. Males homozygous for *fru^F* show very little courtship toward females⁶.

6. P4: *‘Because there were no Mip-GAL4-positive ppk neurons in the uterus (not shown)...’. This result should be shown given its importance.*

It is now included as Supplementary Fig. 4a.

7. P5: *‘re-mating frequency in mated females (Fig. 2c)’... The label above the figure panel states ‘mating’ and not ‘re-mating’. This should be corrected.*

Corrected.

8. P6: *‘We used 27 °C for thermal activation to minimize the activation of the subpopulation of Mip neurons that inhibits mating (i.e., those that express Mip6.0-GAL80, see Fig. 1).’ I do not understand the rationale here. Please explain. Why does*

27 degrees affect these neurons differentially as compared to the non-Gal80 populations?

> To answer this question, we included new data that compares the effect of two temperatures (27 °C and 30 °C) on virgin mating and re-mating of *Mip>dTrpA1*, *Mip6.0>dTrpA1* and *Mip(Mip6.0-GAL80)>dTrpA1* females (see Fig 2d, 2e).

Mip-GAL4 neurons seem to comprise two functionally opposing subsets. This is because thermal activation of *Mip-Gal4* neurons elevates re-mating in mated females, but it also suppresses virgin mating. We generated *Mip^{6.0}-GAL4*, a new *Mip-GAL4* that carries a smaller genomic fragment of the *Mip* promoter than the original *Mip-GAL4*. We then found that activation of *Mip^{6.0}-GAL4* neurons (*Mip^{6.0}>dTrpA1*, 30 °C) almost completely suppresses virgin mating (See Fig. 2d). Remarkably, however, we observed that *Mip^{6.0}>dTrpA1* virgin females incubated at 27 °C remain fully receptive (see Fig. 2d). We compared *Mip^{6.0}>dTrpA1*, *Mip>dTrpA1* and *Mip(Mip^{6.0}-Gal80)>dTrpA1* in 27 °C and 30 °C side-by-side to demonstrate that 27 °C thermal activation partially activates the pro-mating subset without activating the anti-mating subset. Please see the new Fig. 2d and 2e with the explanations about them we have added in this revision.

9. Figure 2e: the result that *Mip* neuron activation enhances mating in the absence of SPR is very interesting. Depending on the effect of *Mip* on SPR signalling (activating or inhibiting (possibly depending on *Mip* levels or cell state), however, this result might be expected although SPR is the receptor for *Mip*. I suggest to include SPR/SPR mutants in the absence of *Mip-TrpA1* and test them at 31 degrees to exclude that the effect of temperature in the background of SPR mutants contributes or explains this effect.

> We performed the suggested experiment, and compared virgin mating in *Spr^{DF/DF}* females at 21 °C and 27 °C. Temperature changes have little impact on the mating rate in SPR-deficient females (see new panel in the upper left of Fig. 3f).

10. Figure 3b: Why was the re-mating assay not carried out for 2 copy *dsx-flp* animals? *Otd-flp* only addresses the brain neurons, but not the additional neurons labelled in the VNC...

> We replaced the old Fig. 3b with a new one. The new one show the results for animals with 2 copies of *dsx^{FLP}* (See Fig. 4b).

11. P8: The authors state: 'We noted that activation of vAL alone does not induce the re-mating phenotype in mated females, whereas activation of all *Mip* neurons does.' I assume they refer to the *dsx-flp* results. If so, this should be made clear. If not, they should name the evidence.

We revised this sentence by inserting "(i.e. *dsx^{FLP}*-positive *Mip-GAL4* neurons)" after "vAL alone" (see lines 293-294)

Reviewer #3 (Remarks to the Author):

This is an interesting study that uses manipulation of specific peptide-containing neurons in *Drosophila* to examine the neural circuitry underlying female sexual receptivity. This field has produced many novel findings and the authors have previously contributed important findings, especially regarding the nature of the peptide ligands and receptors involved. I have several questions and comments that may help improve this manuscript:

Lines 23-25

The summary finishes with the sentence:

“Genetic analyses with a Mip-null mutant suggest that the Mip neuropeptide produced in vAL promotes mating receptivity both in virgins and mated females. But lines 144-145 states that MIP -/- virgins mate normally – so I do not understand how this conclusion (the main one) is reached.

> *Mip* neuron activation promotes mating in virgins and re-mating in mated females. In the *Mip* mutant background, however, the mating-promoting effects of *Mip* neuron activation are largely abolished. Nevertheless, as pointed by the reviewer, the *Mip* gene is dispensable for normal mating receptivity. To clarify these points in the summary, we added “although it is not required for normal virgin mating receptivity” (see line 25)

Line 73

“Silencing of EH, capa and kinin gal4 neurons produces a marginal or non-significant phenotype...” From the Methods, I understand “marginal” to mean $p < 0.05$ or $p < 0.001$ – that sounds significant by every conventional statistical approach I am used to....so what is meant by the term ‘marginal’? non-significant? The authors should explain their thinking on this.

> As seen in Figure S1, the p values for the comparisons between test groups ($GAL4 > UAS-Shi^{fs}$, 30 °C) and parental groups ($GAL4 > +$ or $UAS-Shi^{fs} > +$, 30°C) are higher than 0.05. Thus, we removed “marginal.”

Not sure why MIP-Gal4 > UAS-shi (21 and 30 degrees) in mated females is not reported - please include or explain.

These data are included in the new Fig 1f.

Line 95

The 6 kB gal80 line appears to be a very precise discriminator for presumed ‘inhibitory’ MIP neurons. Was this fragment chosen arbitrarily? Were any others tried? It would be useful to see a schematic map of MIPGAL4+ neurons and MIP-Gal80[6.0]+ neurons. I would have more confidence in the interpretation if the authors had explored the result more with additional constructs/experiments. Were all inhibitory neurons removed? Were any excitatory neurons silenced?

> In addition to the *Mip6.0* fragment, we tested 4 additional constructs with 2–3 kb-

long fragments that tile the 5' genomic region of Mip (see the new Fig. S2 and related explanation in lines 93- 102).

>For the maps of *Mip-GAL4*, *Mip6.0-GAL4*, and *Mip-GAL4/Mip6.0-GAL80*, see Fig. 2a-c and the response to comment 2 of reviewer 1.

Lines 113-114

Please provide a reference for statement regarding egg-laying.

Corrected.

Line 135

The use of tethered SP is useful. Have the authors tried a tethered MIP? That would seem to be a useful reagent in this analysis. What evidence exists to indicate the transmitter phenotype of the SPSN neurons that project from the uterus?

> Indeed, we attempted to generate a membrane-tethered Mip (mMIP) in the same way mSP was constructed. We prepared two mMIP constructs: one with and one without an amidation signal (GRK) at its C-terminus (see slide below). Most neuropeptide precursors have a GRK motif where C-terminal amidation occurs. Unlike with mSP, *ppk* neuron-specific expression of either mMIP or mMIP-GRK has little effect on virgin female mating receptivity (Fig. P3A). We suspect that mMIP is not active because it lacks C-terminal amidation, which is essential for MIP's agonism of SPR⁷. We also noted that neither *ppk*>*mMIP* nor *ppk*>*mMIP-GRK* show anti-Mip staining in *ppk* processes (Fig. P3B). This is consistent with our suspicion that our *UAS-mMIPs* produce mMip lacking C-terminal amidation because the anti-Mip antibody was raised against the C-terminal domain of amidated Mip.

Generation of membrane tethered MIPs (mMIP)

- Membrane tethered domain from gamma-glutamyltransferase (GGT)
- MIP1, AWQSLQSSW-NH₂
- mMIP= GGT(seq)AWQSLQSSW
- mMIP-GRK= GGT(seq)AWQSLQSSWGRK
- Express m-MIPs in the *ppk* neurons that include SPSN

Figure P3 | Generation and testing of membrane tethered Mip (mMIP) constructs.

A. The mating rate for virgin females of the indicated genotypes. mSP-expression in *ppk-Gal4* cells completely suppresses mating, whereas mMip expression has no effect. Here, *ppk-GAL4* drives two copies of *UAS-mMIP* or *UAS-mMIP-GRK*.

B. Anti-Mip or anti-EGFP staining of females of the indicated genotypes. Note that CNSs expressing mMIP or mMIP-GRK in *ppk* neurons do not show additional *ppk*-like anti-Mip staining when compared with anti-Mip staining of control *w¹¹¹⁸*.

Line 152

The text does not reference the correct panels of Figure 2.

> Corrected

Not sure the significance of Figure 2C, as MIP cell activation is supposed to increase receptivity, but in virgin females, it is already near 100% - therefore this experiment appears hampered by a ceiling effect.

> To address this concern, we compared the time course of the cumulative mating rate for several genotypes (See Fig. 3c). This approach allowed us to successfully demonstrate the pro-mating effect of Mip neuron activation in virgin females (Fig. 3e). Nevertheless, Mip overexpression has no discernable effect on the mating rate of virgin females.

Line 162

Unexpectedly, the design switches protocols for thermal activation: previously it used 30 degrees (e.g. Figure 1), now it switches to 27 degrees. The text explains that use of 27 degrees was intended to “minimize the activation” of the inhibitory MIP cells – on what basis was this predicted? How do the authors know the inhibitory neurons require a higher thermal threshold for activation – is there some evidence? Without some plausible explanation, I find this set of experiments very difficult to evaluate.

> See the response to comment 8 of reviewer 2.

Figure 2

In Figure 2b, why not add the Mip-Gal80 to test the contribution of the MIP gene to the “full” complement of activating MIP+ neurons?

> In the limited time given for this revision, it was impossible to combine multiple transgenes with the Mip mutant to perform the suggested experiment. We hope the reviewer can understand our situation.

In Figure 2d, mipGal4>TrpA1 at 27 degrees produces 100% cumulative mating in less than 20 min. However, that genotype at 30 degrees produces ~60% copulation within 1 hr. Aren't these the same behavioral measure, and if so why are the outcomes discrepant? If different behavioral measures, please explain the difference

in the assay, and also why two different assays were used.

> We used the same behavioral assay for two temperatures. For the reasons the results show discrepancies, please see the response to comment 8 of reviewer 2.

Line 192

Silencing Mip/dsx neurons decreased virgin female receptivity (Figure 3a); yet the text states these neurons “maintains female sexual receptivity”. I disagree – at most, the evidence supports the hypothesis that they elevate virgin female receptivity. BTW, 3a uses Kir to silence neurons whereas other experiments use UAS-shi: any explanation for using different reagents to the same end? This adds design variables that lessen the strength of the conclusion regarding neuronal function.

> We noted that temporary silencing of Mip/dsx neurons with Shi^{ts} also markedly suppresses mating rate. The UAS-Kir2.1 data in Fig. 4a has now been replaced by UAS-Shi^{ts} data.

Lines 201-210

The section using confocal microscopy to interpret synaptic interconnections between SPSN and vAL and SAG neurons is not credible. Apparent proximity at the LM level interpreted to represent “reciprocal connections” is not a conservative interpretation.

> The sentence has been removed in the revised manuscript.

Lines 211-250

I was confused by this section in that the text no longer referred to dsx neurons – why not?

To avoid confusion, we revised line 269 by inserting ‘anti-Mip and dsx-positive’ in front of ‘vAL.’

Line 215

EGFP labeled Gal4 neurons are not specified as to age or mating status.

> 3–5-day-old virgin females and males were used. This is now more clear in the materials and methods section (See line 427).

Figure 4a

I do not understand the female/male comparison in this panel. The cartoon does not correspond to the image shown. The male neurons indicated by a star appear by position, number, size and shape to correspond to the vAM of females, but instead are labeled s-vPM. The authors need to clarify their interpretation and present criteria by which they assigned these cell identities. Also what is the point of the star next to the three midline neurons in the male example?

> In the original figure, the male neurons labelled with an asterisk were VMT3 neurons in the metathoracic ganglion, not the AG. Both sexes have VMT3 neurons. To avoid confusion, we have changed to a new male AG image (See Fig. 5a).

Figure 4c

The low power images are of little value without some description. A high power view is offered of the vAM region (I cannot really see the yellow dashed box): How were these two examples (the ones illustrated for us) scored quantitatively in panel e? I think I see 2 cells in the virgin and 2 cells in the mated - is that what the authors scored? In the lateral aspects of T3, the virgin tissue contains 1-2 dark cells and the mated tissue appears to contain these same cells, but the latter are much less intensely stained. That raises the question - is diminished TRIC staining exclusively displayed by vAM neurons? If not, does diminished staining in the vAM retain as much significance? What are the criteria employed here to score a positive cell body? Were all labeled cell groups scored? Is it fair to only compare cell body staining with this method? There appears to be considerable staining of processes. Finally, can the authors demonstrate the validity of the assay (the staining and scoring methods) with some positive control set of neurons, for which there is independent evidence of activity in a given interval? i.e., a positive control.

> In our original manuscript, we scored TRIC-positive and -negative cells according to their relative signal intensity. If the TRIC signal in a given cell body was less than 50% of the average TRIC intensity of its group, we counted it as TRIC-negative. We used this method to compare different groups of neurons with varying basal TRIC signals. *Mip* neurons in the AG showed clear cell-type-specific variations in TRIC-labeling intensity and frequency. For example, *l-vPM* and *vAM* show relatively robust TRIC labeling in all preparations, but *s-vPM* and *s-mPM* produce little to no signal. Although this method allows us to evaluate the differences among cell types, it does not provide separate information about labeling intensity and frequency. To address the concerns raised by the reviewer, we decided to compare the TRIC labeling intensity of cell bodies. This method, however, works well only for cell types that show high and reproducible labeling. It is less suitable for cell types that show low labeling frequency (< 50%) (i.e., *s-vPM*, *s-mPM* and *vAL*). Because we do not count unlabeled cells and the values are presented as an average percentage of a small number of weakly labeled cells, the values are likely overestimates.

In our revision, we compared the TRIC intensities of two groups of *Mip* neurons that show relatively robust signal, such as *vAM* and *l-vPM*. We also examined SAG as a positive control, because like *vAM* neurons, SAG neurons are also active in virgin females and silent after mating. As expected, both *vAM* and SAG show robust changes in TRIC signal intensity before and after mating. *vAM* and SAG neurons have higher TRIC signal before mating than 48 hr after mating.

Ideally, we would be able to include the TRIC signals from the neural processes in the analysis. The TRIC staining panel in Fig. 5c shows SAG somas and processes also show a clear change in TRIC signal after mating. The *Mip* neurons in the AG, however, are packed in a relatively small space, making it difficult to discern the identity of their processes.

Line 245

Typo: 4g should be 4e.

Corrected.

References

1. McRobert, S. P. & Tompkins, L. The effect of transformer, doublesex and intersex mutations on the sexual behavior of *Drosophila melanogaster*. *Genetics* **111**, 89–96 (1985).
2. Ferveur, J. F., Störtkuhl, K. F., Stocker, R. F. & Greenspan, R. J. Genetic feminization of brain structures and changed sexual orientation in male *Drosophila*. *Science* **267**, 902–5 (1995).
3. Anand, A. *et al.* Molecular genetic dissection of the sex-specific and vital functions of the *Drosophila melanogaster* sex determination gene fruitless. *Genetics* **158**, 1569–1595 (2001).
4. Yamamoto, D., Ito, H. & Fujitani, K. Genetic dissection of sexual orientation: behavioral, cellular, and molecular approaches in *Drosophila melanogaster*. *Neurosci. Res.* **26**, 95–107 (1996).
5. Lebo, M. S., Sanders, L. E., Sun, F. & Arbeitman, M. N. Somatic, germline and sex hierarchy regulated gene expression during *Drosophila* metamorphosis. *BMC Genomics* **10**, 80 (2009).
6. Demir, E. & Dickson, B. J. fruitless splicing specifies male courtship behavior in *Drosophila*. *Cell* **121**, 785–794 (2005).
7. Kim, Y.-J. *et al.* MIPs are ancestral ligands for the sex peptide receptor. *Proc. Natl. Acad. Sci. U. S. A.* **107**, 0914764107- (2010).

REVIEWERS' COMMENTS:

Reviewer #1 (Remarks to the Author):

Authors replied all the points that I asked and showed reasonable results. I think this manuscript is ready to be published in Nature communication.

Reviewer #2 (Remarks to the Author):

I am satisfied with the responses and additional experiments carried out by the authors. They have successfully addressed my concerns.

Reviewer #3 (Remarks to the Author):

The revisions are generally effective and greatly improve the manuscript. I commend the authors for a great piece of detective work in deciphering the positive and negative factors in the MIP signals, and in placing the sexually dimorphic vAM and vAL neurons in a defined position in this increasingly detailed circuit.

I have a few concerns about old and new issues as follows: there is no need for additional experiments. Rather I hope textual revisions could help readers more easily follow the complex arguments inherent in this treatment.

My biggest concern is the difficulty I have as a reader (both with the original and revised versions) following the narrative through the difficulty of both MIP activation and inhibition, both virgin receptivity and the re-mating physiology. The Figures move back and forth using either both assays (Figure 1) or just one (Figure 4) or just the other (Figure 5). Combined with considerations of both artificial activation (TrpA1) versus silencing of "normal" activity (shi[ts] or kir). I have to admit to finding it difficult to sustain the basic facts needed to follow the arguments. One suggestion I offer is to revise the Introduction with a better description of the current model for control of female receptivity (if I understand it correctly, SAG neurons promote virgin receptivity and they are inhibited by Mating (via SPSN and SPK signals). Thus virgin receptivity and post-mating physiology are linked. It would have helped me the reader to have this information clearly offered at the outset.

Figure 1.

It is now clear that EH, capa and Kinin were not different from proper secondary controls. Still I think the text is confusing as written. Suggest the following edit:

"Of the 39 GAL4 lines we screened, five exhibited substantial differences when tested at restrictive versus permissive temperatures (Suppl, Figure 1A). However of these five, only two lines —Myoinhibitory peptide (Mip)-GAL4 line and the Diuretic hormone 44 (Dh44)-GAL4 line – continued to exhibit significant differences when compared to proper genetic controls (Supplementary Fig. 1b-f). Here, we focused our analysis on Mip-GAL4, which produces the strongest mating and re-mating phenotypes."

Legend typo – Line 632 (k) should be (m)

Figure 1 typo – Line 153: genotypes marked 1 and 2 appear switched in Figure 1m.

Lines 160 -180 Description of MIP and MIP-6.0 Gal4 lines. There is confusion in these descriptions. Mip Gal4 is expressed in ~230 CNS neurons”, while MIP 6.0 Gal4 is expressed “in 114 brain and 80 VNC neurons” – that equals 224, essentially the same as the MIP-Gal4 number. Further virtually all MIP-Gal4 neurons are MIP-IR, and MIP-6.0 Gal4 “is expressed in most anti-MIP neurons”. If all these equalities hold, why do the schematic maps of Gal4+/antibody- cells appear so different between Fig2 a and b?

It is unfortunate that neither the positive nor inhibitory activities in the MIP-GAL4 could be further defined by the tiling series.

Minor questions regarding Figure 2:

In A, why does CA group # grow from 2 to 4 from left to right schematic?

In general, what is the point of the left schematic, when right schematic is showing the same data plus anti-MIP?

Line 242. “Neither silencing nor activating the Mip/otd neurons affects 9 mating in virgin or mated females (Supplementary Fig. 5e, f). This suggests it is unlikely that the brain Mip neurons are involved. Thus, we concluded that Mip/dsx neurons in the AG constitute a neural circuit that, when active, maintains female sexual receptivity.” This is a negative inference – the data suggests the vAL neurons are necessary but perhaps not sufficient – if so, they would constitute a part of circuit, not all of it.

Mip/dsx female neurons – back and forth between normal activity (inhibiting activation) versus gain of function (dTRPA1). Also back and forth between virgin female and re-mated female effects: these do not always proceed together, yet the text switches back and forth. This is very confusing. Please see my first comment above

1. Line 233 silencing double-positive neurons (hereafter Mip/dsx) markedly suppresses virgin receptivity (Fig. 4a), although their thermal activation does not increase re-mating in mated females (Fig. 4b).
2. Line 244 the brain Mip neurons are involved. Thus, we concluded that Mip/dsx neurons in the AG constitute a neural circuit that, when active, maintains female sexual receptivity.
3. Line 274 Because Mip neuron activation increases re-mating, we reasoned that the relevant Mip neurons would show higher activity levels in virgin females than mated females.

Line 316 When we silenced either the SPSNs or SAGs with Kir2.1 (SPSN- and SAG-, respectively), we found that virgin females are unreceptive.

I am confused here - Why does this happen? – I thought SPSNs signal mating via SP activity to turn off SAGs (Feng et al 2014).

Point-By-Point Responses to reviewer comments: Referee's comments are shown in black, and our responses are shown in blue

Reviewer #3 (Remarks to the Author):

The revisions are generally effective and greatly improve the manuscript. I commend the authors for a great piece of detective work in deciphering the positive and negative factors in the MIP signals, and in placing the sexually dimorphic vAM and vAL neurons in a defined position in this increasingly detailed circuit.

I have a few concerns about old and new issues as follows: there is no need for additional experiments. Rather I hope textual revisions could help readers more easily follow the complex arguments inherent in this treatment.

My biggest concern is the difficulty I have as a reader (both with the original and revised versions) following the narrative through the difficulty of both MIP activation and inhibition, both virgin receptivity and the re-mating physiology. The Figures move back and forth using either both assays (Figure 1) or just one (Figure 4) or just the other (Figure 5). Combined with considerations of both artificial activation (TrpA1) versus silencing of “normal” activity (shi[ts] or kir). I have to admit to finding it difficult to sustain the basic facts needed to follow the arguments.

One suggestion I offer is to revise the Introduction with a better description of the current model for control of female receptivity (if I understand it correctly, SAG neurons promote virgin receptivity and they are inhibited by Mating (via SPSN and SPK signals). Thus virgin receptivity and post-mating physiology are linked. It would have helped me the reader to have this information clearly offered at the outset.

>As suggested, we revised the introduction and inserted following sentences. “The SP and SPR pathway seems inhibitory and it signals the post-mating state by silencing the SPSNs and SAG neurons. Silencing the SPSNs or SAGs renders the virgin females unreceptive to mating, and forced activation of the SAGs overrides the SPSN silencing and makes the females regain the mating receptivity.”

Figure 1.

It is now clear that EH, capa and Kinin were not different from proper secondary controls. Still I think the text is confusing as written. Suggest the following edit:

“Of the 39 GAL4 lines we screened, five exhibited substantial differences when tested at restrictive versus permissive temperatures (Suppl, Figure 1A). However of these five, only two lines —Myoinhibitory peptide (Mip)-GAL4 line and the Diuretic hormone 44 (Dh44)-GAL4 line – continued to exhibit significant differences when compared to proper genetic controls (Supplementary Fig. 1b-f). Here, we focused our analysis on Mip-GAL4, which produces the strongest mating and re-mating phenotypes.”

>Revised as suggested.

Legend typo – Line 632 (k) should be (m)

>Corrected.

Figure 1 typo – Line 153: genotypes marked 1 and 2 appear switched in Figure 1m.

>Corrected.

Lines 160 -180 Description of MIP and MIP-6.0 Gal4 lines. There is confusion in these descriptions. Mip Gal4 is expressed in ~230 CNS neurons”, while MIP 6.0 Gal4 is expressed “in 114 brain and 80 VNC neurons” – that equals 224, essentially the same as the MIP-Gal4 number. Further virtually all MIP-Gal4 neurons are MIP-IR, and MIP-6.0 Gal4 “is expressed in most anti-MIP neurons”. If all these equalities hold, why do the schematic maps of Gal4+/antibody- cells appear so different between Fig2 a and b?

>The referee seems to misunderstand our narration. Due to a technical reason, we were able to examine the overlap between Mip-Gal4 and Mip6.0-Gal4 only indirectly using anti-Mip staining as a reference. Although all anti-Mip neurons express Mip-Gal4, not all Mip-Gal4 is positive for anti-Mip. Actually, 206 Mip-Gal4 neurons are not stained by anti-Mip. Likewise, a significant number of Mip6.0-Gal4 is not positive for anti-Mip. Thus, the overlap between expression patterns of Mip-Gal4 and Mip6.0-Gal4 is significant only in some anti-Mip neurons (30 in the brain and 16 in the VNC), but not in other neurons. We revised the text to clarify this point and avoid potential confusions.

It is unfortunate that neither the positive nor inhibitory activities in the MIP-GAL4 could be further defined by the tiling series.

Minor questions regarding Figure 2:

In A, why does CA group # grow from 2 to 4 from left to right schematic?

>Corrected

In general, what is the point of the left schematic, when right schematic is showing the same data plus anti-MIP?

> The right schematic is sufficient, but we included the left schematic to help readers to appreciate the difference between expression patterns of transgenes more clearly.

Line 242. “Neither silencing nor activating the Mip/otd neurons affects 9 mating in virgin or mated females (Supplementary Fig. 5e, f). This suggests it is unlikely that the brain Mip

neurons are involved. Thus, we concluded that Mip/dsx neurons in the AG constitute a neural circuit that, when active, maintains female sexual receptivity.” This is a negative inference – the data suggests the vAL neurons are necessary but perhaps not sufficient – if so, they would constitute a part of circuit, not all of it.

> We revised as suggested. We inserted “a part of”. Now the text reads as “Thus, we concluded that *Mip/dsx* neurons in the AG constitute a part of the neural circuit that, when active, maintains female sexual receptivity.”

Mip/dsx female neurons – back and forth between normal activity (inhibiting activation) versus gain of function (dTRPA1). Also back and forth between virgin female and re-mated female effects: these do not always proceed together, yet the text switches back and forth. This is very confusing. Please see my first comment above

1. Line 233 silencing double-positive neurons (hereafter *Mip/dsx*) markedly suppresses virgin receptivity (Fig. 4a), although their thermal activation does not increase re-mating in mated females (Fig. 4b).
2. Line 244 the brain *Mip* neurons are involved. Thus, we concluded that *Mip/dsx* neurons in the AG constitute a neural circuit that, when active, maintains female sexual receptivity.
3. Line 274 Because *Mip* neuron activation increases re-mating, we reasoned that the relevant *Mip* neurons would show higher activity levels in virgin females than mated females.

>We revised the introduction as suggested in the first comment above.

Line 316 When we silenced either the SPSNs or SAGs with Kir2.1 (SPSN- and SAG-, respectively), we found that virgin females are unreceptive.

I am confused here - Why does this happen? – I thought SPSNs signal mating via SP activity to turn off SAGs (Feng et al 2014).

> We inserted the following sentence to explain the situation: “This is expected because SP activates an inhibitory G-protein coupled receptor (GPCR) SPR in the SPSNs, which upon activation induces PMR by silencing the SPSNs directly and SAG neurons indirectly.”